# Single-molecule trapping and spectroscopy reveals photophysical heterogeneity of phycobilisomes quenched by Orange Carotenoid Protein

Allison H. Squires [1], Peter D. Dahlberg[1], Haijun Liu [2], Nikki Cecil M. Magdaong[2], Robert E. Blankenship [2] & W.E. Moerner [1]

The Orange Carotenoid Protein (OCP) is a cytosolic photosensor that is responsible for non-photochemical quenching (NPQ) of the light-harvesting process in most cyanobacteria. Upon photoactivation by blue-green light, OCP binds to the phycobilisome antenna complex, providing an excitonic trap to thermally dissipate excess energy. At present, both the binding site and NPQ mechanism of OCP are unknown. Using an Anti-Brownian ELectrokinetic (ABEL) trap, we isolate single phycobilisomes in free solution, both in the presence and absence of activated OCP, to directly determine the photophysics and heterogeneity of OCP-quenched phycobilisomes. Surprisingly, we observe two distinct OCP-quenched states, with lifetimes 0.09 ns (6% of unquenched brightness) and 0.21 ns (11% brightness). Photon-by-photon Monte Carlo simulations of exciton transfer through the phycobilisome suggest that the observed quenched states are kinetically consistent with either two or one bound OCPs, respectively, underscoring an additional mechanism for excitation control in this key photosynthetic unit.

[1] Department of Chemistry, Stanford University, Stanford, CA 94305, USA. [2] Departments of Biology and Chemistry, Washington University in St. Louis, St. Louis, MO 63130, USA. Correspondence and requests for materials should be addressed to W.E.M. (email: wmoerner@stanford.edu)

Photoprotective mechanisms are essential to prevent damage to the photosynthetic apparatus and maintain photo-synthetic efficiency under changing light conditions[1]. One such mechanism is non-photochemical quenching (NPQ)[2,3], where excess energy is thermally dissipated via internal conversion by pigments associated with the light-harvesting antenna complex[4] to reduce production of reactive oxygen species, which can cause irreversible oxidative damage to cells[5,6]. Unlike higher plants, cyanobacteria exhibit a novel NPQ mechanism[7–9] that is triggered, transduced, and actuated by a single water-soluble photoactive protein, known as the Orange Carotenoid Protein (OCP)[8,10–12]. The details of OCP photoactivation, binding, and quenching are of particular interest because they constitute a unique and previously unknown light-activated photoprotective mechanism relevant to an entire class of primary producers in many ecosystems[13–16], and because they may be compatible with, e.g., nanostructured solar technology to offer a biomimetic strategy for light adaptation[17–19].

Cytosolic OCP in its inactive orange form (OCP$^O$) non-covalently binds a ketocarotenoid, 3′ hydroxy-echinenone (3′-hECN[12,20,21]), that bridges the α-helical N-terminal domain (NTD) and the α/β C-terminal domain (CTD), which are joined by a flexible inter-domain linker (Fig. 1a)[12,22]. Under blue-green excitation[23,24], OCP$^O$ is converted into the active red form, OCP$^R$, by the separation of its N- and C-terminal domains[23–26] and translocation of the hECN deeper into the NTD[27,28]. Activated OCP$^R$ (referred to hereafter as simply OCP; the inactive form will be specifically annotated as OCP$^O$) can bind to the phycobilisome (PB), the primary light-harvesting antenna of cyanobacteria, and quenches the excitation energy of the complex (Fig. 1b). Initial in vitro studies indicated that the PB is quenched by one OCP[29] that binds to the core[8,29–33], and that the NTD of OCP is the module responsible for both binding and quenching PBs[34–36]. However, both the exact binding site and quenching mechanism of OCP remain unknown.

A wide array of experimental techniques, including time-resolved spectroscopy[31–33,37], phytological comparison[38], mass spectrometry[35,36], non-linear laser fluorimetry[39], mutagenesis[40–42], computational modeling[43,44], and a single-molecule confocal microscopy study[45], have produced conflicting hypotheses regarding the specific binding site(s) and quenching mechanism of OCP[16,46]. These can be summarized in three fundamental open questions: (1) can OCP bind to multiple structurally and chemically distinct sites on the PBs, or is there a single binding site? (2) Does OCP remain in a single bound conformation, or is binding dynamic and/or heterogeneous? (3) At which pigment or pigments does OCP effect quenching, and is that quenching the result of a structural disruption to the PB, or an energy transfer process involving the OCP carotenoid? The common thread in all three questions is potential structural, conformational, or binding heterogeneity in this system, which if present might be a source of the divergent results reported in the literature.

The PB architecture (Fig. 1c) is assembled primarily from α- and β-subunits of phycobiliproteins with high mutual structural and chemical similarity[47]. In Synechocystis PCC 6803, the PB core comprises three cylinders that each contain four allophycocyanin (Apc) disc-shaped αβ trimers, most of which emit at 660 nm. The two discs at antiparallel ends of each of the two bottom cylinders together contain three subunit variants (ApcD, ApcE, and ApcF), which form the site of energy transfer to PSI and PSII in vivo via red-shifted pigments emitting at 680 nm located in the ApcD/ApcE subunits, respectively[48,49]. Excitons are funneled into the PB core by six C-phycocyanin rods[50], which in the wild-type Synechocystis PCC 6803 each consist of three stacked C-phycocyanin hexamers. We used the truncated CB-PB mutant of Synechocystis PCC 6803[51] to ensure structural homogeneity

(see Methods for details). The literature consistently reports that CB-PB is effectively quenched by OCP[29,33]. This mutant is useful because it has been previously suggested that both the architecture and the spectroscopic properties of phycobilisomes, especially the C-phycocyanin rods, may be dynamic and/or modular[47,52–54], facilitating structural remodeling in vivo[54,55].

Here, we employ single-molecule photophysical techniques to directly detect heterogeneity in OCP-related NPQ in phycobilisomes. Using an Anti-Brownian ELectrokinetic (ABEL) trap[56–59] to isolate single intact PB antenna complexes with and without OCP in free solution for seconds or longer, we describe single-particle spectroscopic observation of both unquenched and quenched states of energy transfer. In agreement with previous reports, we observe only excitation-dependent spectroscopic heterogeneity in unquenched PBs[60], with relatively little global structural heterogeneity. Surprisingly, in the presence of OCP, two photophysically distinct and stable quenched populations are observed. Simulated brightness and fluorescence lifetimes for CB-PB with different combinations of possible OCP quenching sites, produced using a multi-compartment model of the phycobilisome from the literature[61] that was based on ultrafast transient absorption and emission experiments[31–33], closely reproduce the experimentally observed populations only under the key assumption that the two quenched populations are the result of one (state $Q_1$) and two (state $Q_2$) OCPs bound to the PB, respectively.

## Results

**Bulk spectroscopy of OCP-quenched phycobilisomes.** For direct comparison to our single-molecule measurements, we first determined the bulk emission spectra and fluorescence lifetime decays for unquenched CB-PB and fully quenched[29] OCP + CB-PB (40:1 OCP:CB-PB), as described in Methods. Based on our bulk fluorescence measurements (Fig. 1d), the fully quenched OCP + CB-PB emits only ~9% of the fluorescence of unquenched CB-PB (see Supplementary Note 1). The peak-normalized emission spectra show a clear blue shift in the quenched state that corresponds to a decrease in the center of mass of the emission spectrum, $\lambda_{CM}$, from 669.6 nm (unquenched) to 665.6 nm (quenched), indicating that a lower proportion of total fluorescence is emitted from the quenched core, while a higher proportion is emitted from the C-PC hexamer rods. Here, we employed minimal model fits to bulk fluorescence lifetime decays for unquenched CB-PB (1-exponential) and fully quenched OCP + CB-PB (2-exponential), shown in Fig. 1e. The longer lifetime component (1.5 ns) for the quenched OCP + CB-PB complex is very similar to the unquenched lifetime (1.6 ns), while the shorter lifetime component for the quenched complex is only 0.1 ns. From both the emission and lifetime data, we calculate that most complexes are quenched (>95%; see Supplementary Note 1).

**Single complexes isolated in the ABEL trap.** The ABEL trap confines single particles in solution, permitting the measurement of their photophysical states as illustrated by Fig. 2a[56–59]. At low pM concentrations, a single particle enters the trapping region of a microfluidic cell, within which its position may be monitored. Through fast electrophoretic or electroosmotic feedback, its Brownian motion may be counteracted to confine it within the center of the trapping region for extended periods of time. Simultaneously, fluorescence brightness (Br), polarization (FPol), lifetime (τ), and emission spectrum (Em(λ), also described by the spectral center of mass, $\lambda_{CM}$) are monitored for each trapped particle. The ABEL trap is well suited to characterize individual phycobiliproteins[62–64] and other photosynthetic protein complexes[65–67] because it does not require analyte

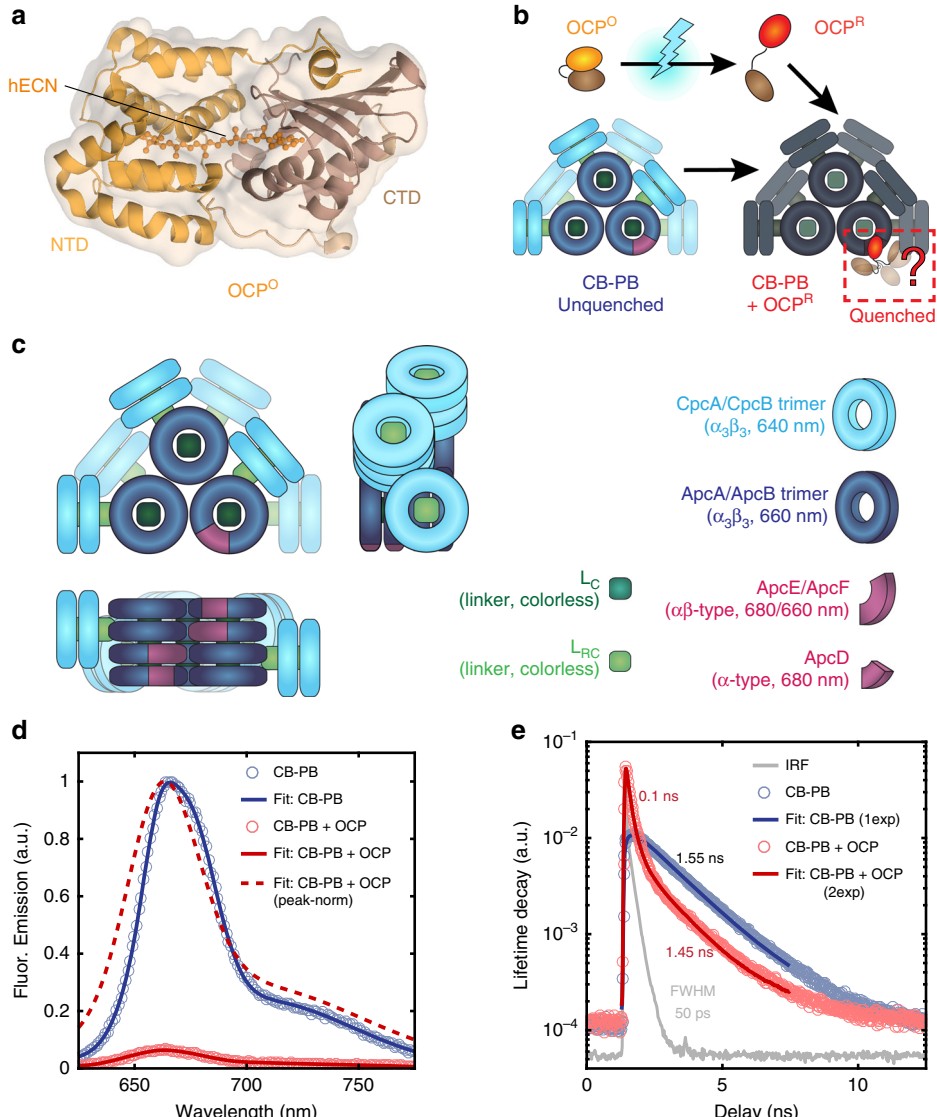

**Fig. 1** OCP quenches fluorescence in the phycobilisome. **a** Structure of OCP in inactive orange form, from PDB 3MG1[42]. A keto-carotenoid spans the interior of the N- and C-terminal domains (NTD and CTD, respectively), which are connected with a flexible linker and separate upon photoactivation. **b** Schematic of OCP activation. Upon absorption of blue-green light, $OCP^O$ is converted to $OCP^R$, which binds to and quenches an unknown location on the core of the phycobilisome (PB). **c** Structure of the truncated PB mutant used in this study (CB-PB from *Synechocystis* PCC 6803[51]). Disc-shaped trimers of phycobiliproteins are joined by linker proteins into stacks that make up the core and rods of the PB. **d** Bulk fluorescence emission spectrum of PB in the absence (navy) and presence (maroon) of bound $OCP^R$. Solid lines: area-normalized for total relative fluorescence intensity. Dotted line: peak-normalized. **e** Bulk lifetime decays for PB in the absence (light blue) and presence (light red) of bound $OCP^R$. Single- and double-exponential fits are shown in navy and maroon, respectively. Gray: instrument response function, FWHM 50 ps

immobilization, which might perturb the spectroscopic properties or energy transfer in these delicate structures (see Supplementary Note 2 for additional discussion).

Here, we examined both single CB-PB phycobilisomes and single pre-activated and pre-bound OCP + CB-PB complexes in the ABEL trap. A raw data trace in these variables for several consecutively trapped single unquenched CB-PBs is shown in Fig. 2b. Each trapping event is highlighted in blue. Because each event consists of a single level in every photophysical parameter, and because the combination of parameters observed from these events falls into a single population (vide infra), we consider them to belong to a single photophysical state, which we will refer to as the unquenched state, or state U. Each event here is labeled as state U (top), and the mean photophysical parameters for state U in each measured variable are indicated with blue dotted lines and labeled at right. The consistency of the photophysical state

observed during trapping for each particle, and also across the population of trapped particles, indicates that the CB-PB assemblies are intact with a homogeneous structure (within measurement error), and do not undergo measurable photo-damage or photodynamics while trapped using this range of excitation power (here: power 160 nW, or intensity 4.8 W cm$^{-2}$; see also Supplementary Figure 1). As has been previously reported, at higher excitation powers (here: >20 W cm$^{-2}$), laser-induced photodamage and photodynamics are readily observable for the unquenched phycobilisome[60] (Supplementary Figures 2 and 3).

A similar raw data trapping trace for several individual quenched OCP + CB-PB complexes can be seen in Fig. 2c (excitation: 15 W cm$^{-2}$). Higher powers were used to study quenched complexes because they are an order of magnitude (or more) dimmer than the unquenched phycobilisomes. To achieve sufficient signal to noise, these higher powers were necessary. The

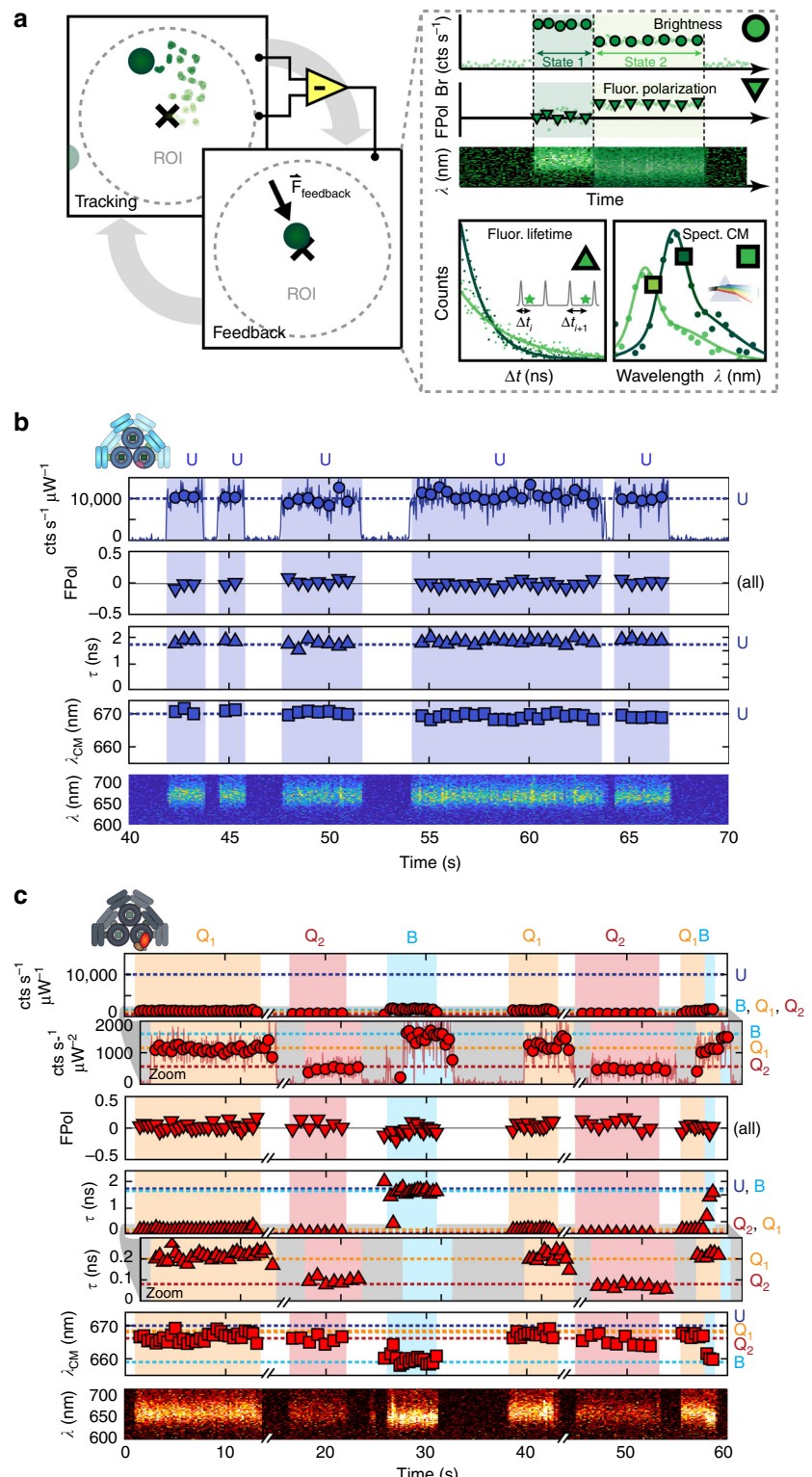

**Fig. 2** Anti-Brownian trapping of single unquenched and quenched phycobilisomes. **a** ABEL trap schematic operation and multiparametric data collection. The position of a single particle diffusing within an observation region of interest (ROI) is monitored, while feedback is applied in an active control loop to maintain the particle within the ROI (left). Multiple photophysical parameters are monitored for each trapped particle in real time, including brightness (Br), fluorescence polarization (FPol), fluorescence lifetime ($\tau$), and emission spectrum (Em($\lambda$)), which can be reduced to its center of mass, $\lambda_{CM}$. **b** Raw trapping data for all parameters (Br, FPol, $\tau$, $\lambda_{CM}$, and Em($\lambda$)) for the CB-PB complex in the absence of OCP (incident excitation intensity 4.8 W cm$^{-2}$) shows a consistent photophysical state, U. Data are plotted in either 800-photon groups (markers) or 30-ms bins. **c** Raw trapping data for all parameters for CB-PB quenched by OCP (1:40). At least three distinct photophysical states (B, $Q_1$, and $Q_2$) are evident in the sample containing the quenched complex. Data are plotted in either 500-photon groups (markers) or 30-ms bins

**Table 1 Extracted photophysical parameters for each state**

| State | Br (cts s$^{-1}$ $\mu$W$^{-1}$) | $\tau$ (ns) | $\lambda_{CM}$ (nm) | FPol |
|---|---|---|---|---|
| U | 10420 ± 1670 | 1.66 ± 0.14 | 669 ± 1.4 | 0.0 ± 0.05 |
| Q$_1$ | 1160 ± 250 | 0.21 ± 0.03 | 667 ± 1.0 | 0.0 ± 0.05 |
| Q$_2$ | 600 ± 140 | 0.09 ± 0.02 | 665 ± 0.9 | 0.0 ± 0.06 |
| B | 1630 ± 320 | 1.64 ± 0.14 | 659 ± 0.9 | 0.0 ± 0.05 |

Extracted parameters {Br, $\tau$, $\lambda_{CM}$, and FPol} and standard deviation of each observed photophysical state {U, Q$_1$, Q$_2$, and B}, as determined by Gaussian fits to the aggregated data for each state shown in Supplementary Figures 5–8. Errors represent the standard deviation of each Gaussian fit

trapped quenched complexes are clearly photophysically stable at these powers, as shown by the long, single-level trapping events (see Fig. 2 and Supplementary Figure 4 for sample trapping events).

Two additional zoomed-in views are provided in Fig. 2c for the fluorescence brightness (Br) and lifetime ($\tau$) panels to more clearly show the differences among states B, Q$_1$, and Q$_2$ in those variables. It is evident from the decreased brightness alone that these states may be quenched. However, there is also significant heterogeneity in Br, $\tau$, and Em($\lambda$) among individual trapping events. The three unique combinations of photophysical parameters observed for the sample containing the quenched complex will be referred to as the blue state (B) and two quenched states (brighter: Q$_1$ and dimmer: Q$_2$). Each of these states is highlighted in a different color within each trapping event and labeled above, and the mean photophysical parameters for each state are indicated with dotted lines and labeled at right (B: light blue, Q$_1$: orange, Q$_2$: red). The extracted parameters for state U, which is not observed in this data trace, are also included at right for reference. Possible transitions among states[45], as seen in the final example event in this trace, are observed only rarely, particularly at low concentration, and may be caused by stochastic replacement of a trapped particle with another particle that enters the trapping area. (See Supplementary Figure 4 for an extended raw data trace showing many trapping events in Br and $\tau$, and Supplementary Note 3 for additional discussion of possible transitions.)

**Experimentally determined photophysical states**. To establish the number and characteristic photophysical parameters of the states described above, we aggregated many such trapping events for both unquenched and OCP-quenched CB-PB samples. Figure 3a, b show scatter heatmaps of trapping data sets in the Br-$\tau$, Br-$\lambda_{CM}$, and Br-FPol projections for unquenched CB-PB (92 trapping events) and quenched CB-PB + OCP (224 trapping events), respectively, where each point represents the photophysical state of a group of 200 consecutive photons and is colored according to the local density of such points, as detailed in Methods. As expected from the time-resolved data, one dominant state (U) is observed for the unquenched CB-PB. At least three possible states are observed for quenched CB-PB + OCP: two that are dim with short lifetimes (states Q$_1$ and Q$_2$; see inset for the Br-$\tau$ projection in the left panel of Fig. 3b), and a third state (state B) that is dim and blue-shifted with an unchanged lifetime relative to the unquenched CB-PB. Both the quenched and unquenched phycobilisomes exhibit nearly complete depolarization (FPol ≈ 0), indicating that substantial energy transfer among many non-aligned dipoles occurs. The mean values and standard deviations for each of these states were determined for each parameter via Gaussian fits (see Supplementary Figures 5–8), and are presented in Table 1.

Figure 3c shows a parameter map in each of the same projections as the panels above (from left to right: Br-$\tau$, Br-$\lambda_{CM}$,

and Br-FPol) with the most likely values for each state plotted inside 95% confidence ellipses (U: navy, B: light blue, Q$_1$: orange, Q$_2$: red). The collective emission spectra for each of these four states are shown in Fig. 3d). As expected, Em($\lambda$) measured in the ABEL trap for unquenched single CB-PB particles matches the bulk unquenched spectrum from Fig. 1d. The emission spectra for Q$_1$ and Q$_2$ are both qualitatively consistent with quenching by OCP; when fluorescence emission from the core is quenched, a higher proportion of the total emitted fluorescence comes from the C-PC hexamers. The emission spectrum of state B is substantially blue-shifted, consistent with a single decoupled C-PC hexamer (vide infra).

Figure 3e shows the collective fluorescence lifetime decays with fits for each of the four states, confirming that $\tau_U$ is consistent with $\tau_{bulk}$ for the unquenched state. The lifetime decay for state B is consistent with the expected lifetime decay for C-PC hexamers and energetically decoupled rods (see Supplementary Figure 9 for C-PC rod sample data, CpcG2-PBS, taken for rods and hexamers in the ABEL trap), while the quenched state lifetimes, $\tau_{Q1}$ and $\tau_{Q2}$, are significantly shorter and mutually distinct, and match the values derived from the 200-photon groups shown in the scatter heatmaps above and in Table 1. The lifetime decays for all four populations are well fit by single exponential decay models, in contrast to the bulk quenched case (Fig. 1e).

**Time dependence of quenched state populations**. Comparison of data sets taken over the course of 5 h using a single OCP + CB-PB sample diluted to a concentration of 10 pM at time $t = 0$ clearly show that the relative proportion of states Q$_1$ and Q$_2$ changes over time. Figure 4a shows three zoomed-in scatter heatmaps in the Br-$\tau$ projection for data collected at $t \leq 10$ min (left; 5 min data collection, $N = 72$ events), $t = 50$–65 min (center; 15 min data collection, $N = 198$), and $t = 285$–310 min (right; 25 min data collection, $N = 331$ events) after dilution. While most of the observed photophysical states for these 200-photon groups are initially in state Q$_2$ (left), after 60 min the Q$_1$ population has significantly increased (center), and after 5 h the majority of 200-photon groups are in state Q$_1$ (right). Data from these and many additional time points are plotted in Fig. 4b, which shows the proportion of Q$_1$ or Q$_2$ levels observed in a 5-min data set relative to the total number of quenched levels: $N_{Q(1 \text{ or } 2)}/(N_{Q1} + N_{Q2})$. Here, all consecutive 200-photon points that fall in the 95% confidence ellipse of the same state are considered to be a single level. At $t = 190$ min, the sample in the microfluidic cell was exchanged for additional sample from the original dilution due to reduced overall capture rate into the trap. These data clearly show that the proportion of observed Q$_2$ levels decreases over the course of several hours, while the proportion of Q$_1$ levels rises.

We also observed more unquenched complexes with increasing time. Figure 4c shows the proportion of trapped unquenched complexes (U) relative to the total number of U, Q$_1$, and Q$_2$ observed: $N_U/(N_{Q1} + N_{Q2} + N_U)$ as a function of time. Conservatively, any trapping level that exhibits one or more 200-photon groups with a brightness above 5000 cts s$^{-1}$ $\mu$W$^{-1}$ are included in $N_U$, while $N_{Q1}$ and $N_{Q2}$ are counted as described above. The increase in unquenched complexes over time suggests that over the course of several hours, OCP unbinds from CB-PB under our experimental conditions (Q$_1$ to U). Unbinding of OCP from the core could also explain the transition from state Q$_2$ to Q$_1$ over time under the assumption that Q$_2$ is caused by the presence of a higher number of bound OCPs than Q$_1$.

**Discussion**
Of the three states observed for the quenched OCP + CB-PB complex, state B is least consistent with the expected changes in

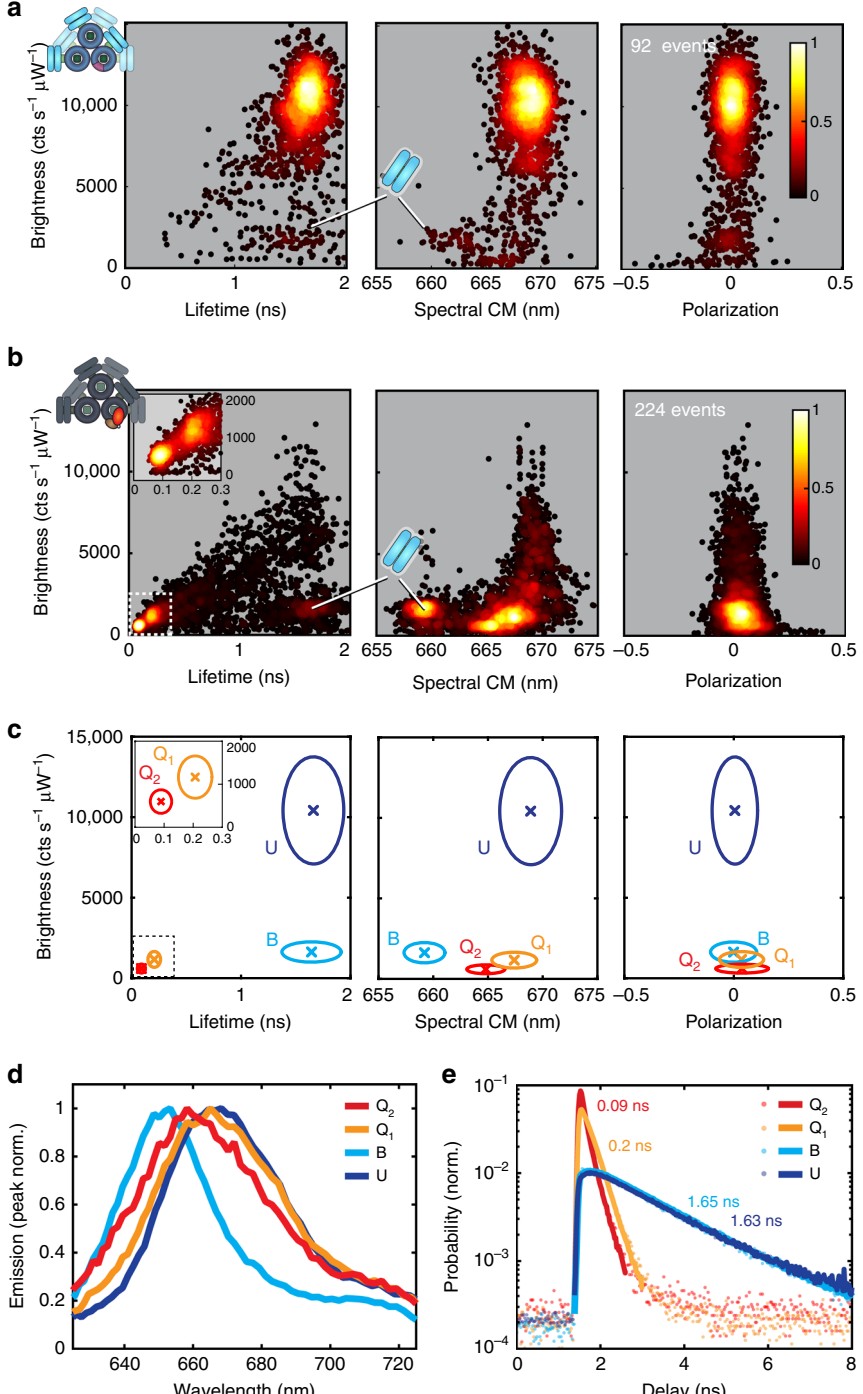

**Fig. 3** Photophysical states of unquenched and OCP-quenched CB-PB Phycobilisomes. **a** Scatter heatmap of the photophysical state (U) observed for the unquenched CB-PB phycobilisome, shown in Br-$\tau$, Br-$\lambda_{CM}$, and Br-FPol projections (left, right, and center, respectively). Each point represents 500 photons, colored according to the normalized local density of points for each panel (see colorbar). The location of state B, very rarely observed here, is also indicated for reference. **b** Scatter heatmap of photophysical states (B, $Q_1$, and $Q_2$) observed for the sample containing OCP-quenched CB-PB phycobilisome, shown in Br-$\tau$, Br-$\lambda_{CM}$, and Br-FPol projections (left, right, and center, respectively). Each point represents 500 photons, colored according to the normalized local density of points for each panel (see colorbar). **c** Summary map of observed photophysical states U, B, $Q_1$, and $Q_2$, each depicted by the population mean (x) and 95% confidence ellipse. **d** Fluorescence emission spectra associated with each photophysical state. **e** Collective fluorescence lifetime decay and single-exponential maximum likelihood fit for each photophysical state

photophysical parameters that would be induced by the presence of a quencher, and therefore we will briefly discuss it here before turning to the question of the two quenched states, $Q_1$ and $Q_2$: Spectroscopically, state B appears to be most consistent with

fluorescence from a single C-PC hexamer for three reasons. First, its spectrum is significantly blue-shifted compared to APC emission. Second, its long lifetime indicates that the trapped object is not being quenched by OCP, and the lifetime decays of

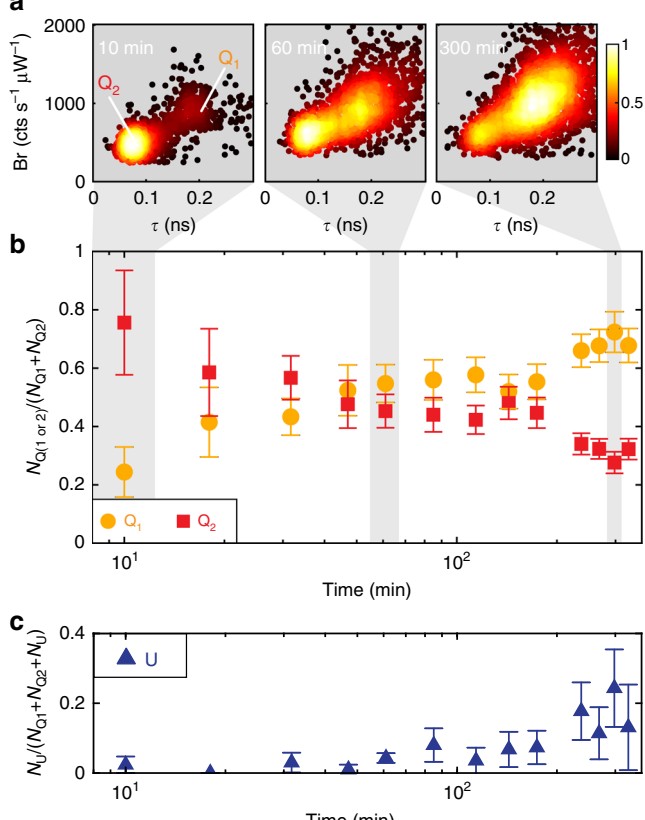

**Fig. 4** Change in relative $Q_1$, $Q_2$, and U populations over time. **a** Scatter heatmaps of brightness vs. fluorescence lifetime taken 10 min (left), 60 min (center), and 300 min (right) after sample dilution to 10 pM. Each point represents 200 photons, colored according to the normalized local density of points for each panel (see colorbar). **b** Relative fraction of $Q_1$ (orange) and $Q_2$ (red) trapping events as a function of experiment time from 10 min up to 5 h after sample dilution to 10 pM of CB-PB. Points highlighted in gray are associated with the corresponding scatter heatmaps shown in **a**. **c** Relative fraction of unquenched (state U) trapping events observed as a function of time from 10 min up to 5 h after sample dilution to 10 pM of CB-PB. Error bars in **b** and **c** represent standard deviations as calculated by error propagation

state B levels show no evidence of the presence of additional, shorter lifetime components (see Supplementary Figure 10). Finally, we trapped C-PC rods from the ΔAB mutant of *Synechocystis* PCC 6803[68] for comparison with state B: the distribution of brightness levels observed for trapped C-PC rods (Supplementary Figure 9), which should each contain ~3 hexamers on average, can be fit under the assumption that the rods are dissociating in the solution and the constraint that $Br_{3Hex} = 3 \times Br_{1Hex}$ and $Br_{2Hex} = 2 \times Br_{1Hex}$ to give the expected brightness and standard deviation for a single C-PC hexamer as $Br_{1Hex} = 1500 \pm 400$ cts s$^{-1}$ μW$^{-1}$. This value is very similar to the observed brightness of state B ($Br_B = 1630 \pm 320$ cts s$^{-1}$ μW$^{-1}$) for the quenched OCP + CB-PB sample. As noted above, the observed lifetime for state B, $\tau_B = 1.64 \pm 0.14$ ns, also matches the lifetime of C-PC hexamers and partially decoupled rods (Supplementary Figure 9). We therefore conclude that state B must represent single C-PC hexamers that have detached from a CB-PB.

Both states $Q_1$ and $Q_2$ are qualitatively consistent with the expected photophysical characteristics of an OCP-quenched CB-PB complex. They are dim and slightly blue-shifted, with short lifetimes, indicating the presence of a competing relaxation pathway for excitons in the PB core that might otherwise produce

fluorescence in an unquenched phycobilisome. Several types of explanations might account for the existence of these two spectroscopically distinct quenched states of the phycobilisome, all of which have implications for the likely binding site(s) and quenching mechanism of OCP:

One class of explanations assumes that quenching is effected by only one OCP bound to one specific site on the phycobilisome core, and therefore any observed spectroscopic heterogeneity must arise from differences in either the physical or electronic structure of the OCP + CB-PB complex. However, there is no evidence of substantial spectroscopic or physical heterogeneity in the absence of OCP (state U), and the observed spectroscopic parameters of states $Q_1$ and $Q_2$ are consistent with increasingly strong quenching at the core of an intact CB-PB rather than with a substantial change to the physical organization and structure of the phycobilisome. For example, although removing one or more C-PC hexamers from the core might produce a dimmer state, the emission would then red-shift rather than blue-shift from $Q_1$ to $Q_2$. Furthermore, this view is incompatible with the observation that the quenched complexes shift from the more-quenched $Q_2$ state to the less-quenched $Q_1$ state over time.

The possibility of a subtle conformational shift that produces a large change in electronic structure between $Q_1$ and $Q_2$, or a variation on this explanation where one state is the true quenched state and the other is an intermediate, seems initially plausible. However, we observe that the quenched CB-PB + OCP complex remains in either $Q_1$ or $Q_2$ during the vast majority of quenched trapped events, and does not typically transition between them. For this to be possible, the conformational shift distinguishing $Q_1$ from $Q_2$ would have to be highly stable over many seconds or longer, and would have to decrease the equilibration time for excitons moving within the phycobilisome while simultaneously making the complex dimmer. This might be possible if, for example, the rate of energy transfer toward the site of quenching were significantly increased in $Q_2$ as compared to $Q_1$. A related possibility is that OCP itself can exhibit variations (for example, by oligomerization, conformational changes, or changes in electronic structure) that create precisely two different quenching efficiencies. To test these hypotheses, we simulated the effect of a single bound OCP with variable quenching strength, as described below.

Another set of hypotheses arises under the assumption that only one OCP at a time binds to cause quenching, but that multiple spectroscopically distinct binding sites are available on the core. This line of thinking holds some appeal due to the inherent structural similarities of the various APC subunits that make up the core cylinders, which might provide structurally similar binding sites for OCP. It is also potentially compatible with multiple possible quenching mechanisms that have been previously suggested, for example that quenching is caused by structural disruption of the phycobilisome upon OCP binding, or that hECN is involved in quenching one or more proximal chromophores upon binding, either through coupled energy transfer[30,69,70], or through charge transfer[31,33]. In this scenario, it is even possible to describe a system that is consistent with the population shift that we observe from $Q_2$ to $Q_1$ over time, for example if the site that produces strong quenching is kinetically favored during binding, but less energetically stable than the site that produces weaker quenching. However, this requires precisely two types of available binding sites with similar stability but very different quenching efficiency, which we find to be less likely than the simpler alternative presented below.

The phycobilisome core is not only composed of highly similar APC subunits, but it also possesses an overall $C_2$ symmetry due to the two antiparallel bottom cylinders, which have

identical subunit composition. Therefore, it is reasonable to suppose that two identical, $C_2$-symmetric binding sites for OCP might exist on the phycobilisome core. In this case, it must be assumed that binding of one OCP to the core does not substantially disrupt the physical structure of the phycobilisome, since a second OCP must be able to bind in a symmetric position. $Q_1$ might then correspond to a single bound OCP quenching the phycobilisome, while $Q_2$ would correspond to two bound OCPs that in tandem produce even stronger quenching. This is also consistent with our observation that (under saturated initial binding conditions) upon dilution to extremely low concentration, the system is out of equilibrium, and therefore slowly shifts from $Q_2$ toward $Q_1$ (and towards unquenched phycobilisomes) as OCPs stochastically unbind over time.

This hypothesis not only provides the simplest logical explanation of the observed $Q_1$ and $Q_2$ states, but is also in agreement with the current understanding of energy transfer in the phycobilisome. To show this, we have directly compared our experimental results with the predictions of a compartmental computational model of energy transfer in the *Synechocystis* PCC 6803 phycobilisome, which was developed by Berera and co-workers[61]. This model is based on extensive ultrafast absorption and emission spectroscopy data for the full phycobilisome, as well as for a core-only mutant and for the CB-PB mutant used in this study, and also includes data for complexes quenched by OCP[31–33]. Based upon this CB-PB model, we have created photon-by-photon Markov chain simulations to predict the spectroscopic states that we would expect to observe for a variety of possible quenching scenarios.

Figure 5a shows the overall compartment architecture and connectivity of the CB-PB model employed here, with the various possible quenching locations for OCP on the phycobilisome core labeled *a*–*f*. Generally, each compartment consists of a C-PC or APC hexamer that is connected to each of its closest geometric neighbors. The exceptions to this are compartments *a* and *d*, which represent only the $APC_{660}$ portions of the hexamers that contain the terminal pigments on each of the two bottom cylinders, and compartments *a'* and *d'*, which represent the $APC_{680}$ subunits (ApcD, ApcE, and ApcF) from those hexamers. The forward and backward rates of exciton transfer between each pair of connected compartments were based directly on the published model for the CB-PB mutant, and are detailed in Supplementary Table 1. Absorption probabilities for each compartment are given in Supplementary Table 2. To simulate different possibilities for OCP binding and quenching, we varied only the model quenching connectivity—that is, which compartment is quenched by a bound OCP—and the forward transfer rate from that compartment to the OCP. This differs slightly from the approach employed by Berera et al., who modeled OCP quenching occurring simultaneously for all compartments within the core at a rate of $k_Q = 54$ ns$^{-1}$.

Figure 5b shows a scatter plot in the Br-$\tau$ projection of the predicted photophysical parameters for states U, B, $Q_1$, and $Q_2$ obtained from this model. Each point represents 200 simulated photon emissions. Here, the absorption rate of the CB-PB model complex was tuned so that the brightness of state U matches the brightness that we observe in the ABEL trap, and all other simulations use that same scaling. State B was modeled as a single, unconnected C-PC hexamer, and its brightness, lifetime, and emission spectrum predicted by the simulation are nearly identical to those experimentally observed for state B, confirming our assignment of state B as an uncoupled C-PC hexamer.

To settle on a model for $Q_1$ and $Q_2$, it was necessary to test the different possibilities for OCP quenching as discussed above. First, we asked whether or not $Q_1$ and $Q_2$ could result from a single

bound quencher located at different quenching sites, or with different quenching strength due to oligomerization or structural variations. Figure 5c shows the predicted brightness and lifetime for the four symmetrically unique cases of a single bound OCP (located at compartment *a*, *a'*, *c*, or *e*) as a function of the forward transfer rate, $k_Q$, from that compartment to the OCP. Even at the highest quenching rates tested, the expected brightness and lifetime do not reach the $Q_2$ state. Interestingly, the maximum achievable predicted quenching for one bound OCP, regardless of the site, is very similar to our observed state $Q_1$.

In contrast, modeling two bound OCPs easily reproduces the spectroscopic characteristics of state $Q_2$. Figure 5d shows the predicted brightness and lifetime for the four unique combinations of two rotationally symmetric binding sites for two bound OCPs (*ad*, *a'd'*, *bc*, and *ef*; see Supplementary Note 4 and Supplementary Figure 11 for consideration of alternative combinations of sites) as a function of increasing quenching rate (which was assumed to be equal for both OCPs). The only combination that does not achieve the $Q_2$ state, even at the highest quenching rates, is the *ef* combination, which represents OCPs bound at either end of the top core cylinder. The other three combinations (*ad*, *a'd'*, and *bc*) all asymptotically approach spectroscopic parameters that are very similar to $Q_2$ at high quenching strength. For the purposes of the results shown in Fig. 5b, we have taken the effective quenching rate for an individual compartment to be 540 ns$^{-1}$, because this is roughly the minimum rate required in our model to match the experimentally observed parameters for both states $Q_1$ and $Q_2$. These results imply that binding at any pair of antiparallel sites of the bottom two cylinders in the core could produce sufficient quenching to explain state $Q_2$. This is consistent with previous suggestions that OCP binds the basal rods[38], that it might quench either $Apc_{660}$[31,33] and/or $Apc_{680}$[39], and that it may bind near ApcE[35,36].

Individual OCP-quenched phycobilisomes spectroscopically profiled using the ABEL trap each clearly represent one of two distinct photophysical states. One of these states is quenched to ~11% of the brightness of an unquenched CB-PB, with a fluorescence lifetime of 0.21 ns ($Q_1$). The second is more strongly quenched to ~6% of the unquenched CB-PB brightness, with a fluorescence lifetime of 0.09 ns ($Q_2$). At binding saturation, most complexes were initially in the $Q_2$ state, but over the course of several hours we observed increasing populations of $Q_1$ and unquenched CB-PB complexes, which we attribute to unbinding of OCP from the phycobilisome at the low concentrations (pM) used in the ABEL trap. These results suggest that two OCP molecules can simultaneously bind to the phycobilisome core to generate the $Q_2$ quenched state, and that the less-quenched $Q_1$ state is observed when only one OCP is bound. This hypothesis and the photophysical parameters that we experimentally measured for $Q_1$ and $Q_2$ are in good agreement with the predictions of a simple compartmental model of the phycobilisome.

The observation of two binding sites on the CB-PB core for OCP and the direct characterization of their spectroscopic parameters and stability narrows down the possible range of structural and quenching interactions of OCP with the phycobilisome. First, it is very likely that the two binding sites that produce states $Q_1$ and $Q_2$ are $C_2$-symmetric, and therefore only one chemically and structurally unique type of site on the phycobilisome is quenched in the presence of OCP. Second, both quenched states are mutually photophysically stable; we did not observe photodynamics caused by e.g.: transitions between $Q_1$ and $Q_2$ or to or from the unquenched state, and the measured spectroscopic parameters for each state remained stable during trapping. Thus the bound state of OCP must be energetically stable relative to any binding and unbinding intermediates that might exist. These

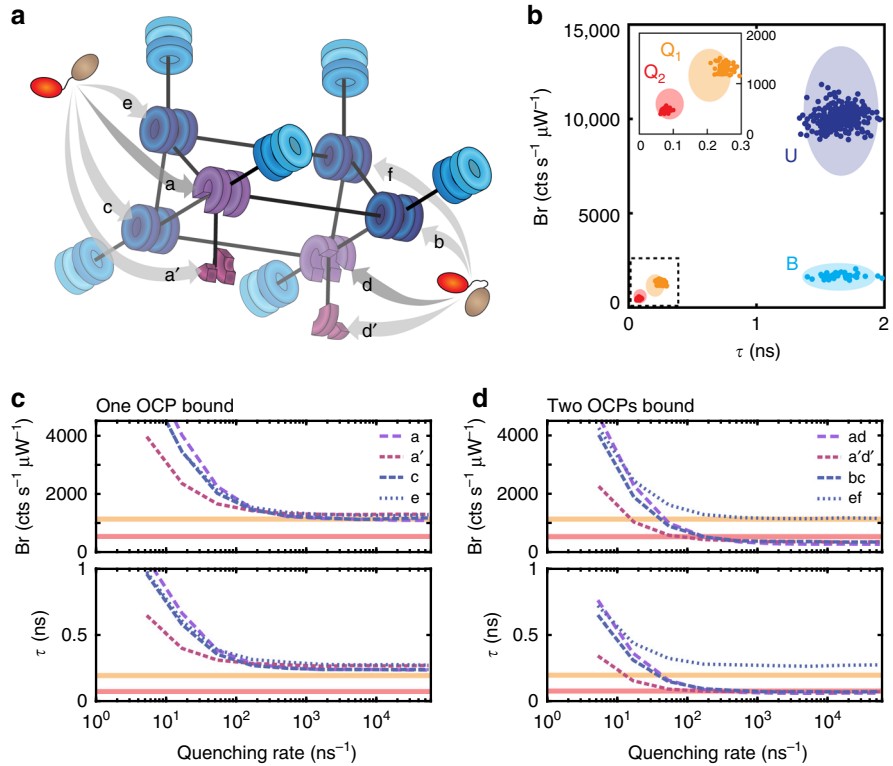

**Fig. 5** Compartmental model of OCP-quenched CB-PB phycobilisome for one and two quenchers. **a** Schematic structure and connectivity of compartmental CB-PB model, after van Stokkum et al.[61]. All components are colored and positioned according to the structure shown in Fig. 1. Eight possible quenching sites for OCP are labeled (*a–f*, plus *a'* and *d'*). **b** Scatter plot of brightness vs. fluorescence lifetime simulation results for four model conditions: unquenched (U), C-PC hexamer only (B), one OCP bound at site *a*, *a'*, or *c* ($Q_1$), or two OCPs bound at site combinations *ad*, *a'd'*, or *bc* ($Q_2$), with an assumed quenching rate of 540 ns$^{-1}$ for each OCP compartment. Ellipses show the locations of the 95% confidence intervals for the four experimentally measured states U, B, $Q_1$, and $Q_2$, as originally depicted in Fig. 3c. **c** Brightness and fluorescence lifetime of the CB-PB phycobilisome for the case of one bound quencher located at each of the four rotationally unique sites *a*, *a'*, *c*, and *e*, plotted for increasing quencher strength. **d** Brightness and fluorescence lifetime of the CB-PB phycobilisome for two bound quenchers at each of the four unique pairs of $C_2$-symmetric sites *ad*, *a'd'*, *bc*, and *ef*, plotted for increasing quencher strength. Horizontal lines in **c** and **d** show the experimentally measured parameters for $Q_1$ (orange) and $Q_2$ (red)

data also show that if multiple bound conformations of OCP exist, as has been suggested in particular for the C-terminal domain, these conformations do not produce spectroscopically distinct quenched states on timescales relevant to the measurements described here (between 1 and $10^4$ ms).

The compartmental model employed in this work is a rough approximation of energy transfer through the phycobilisome structure and to OCP, so we note only that the range of energy transfer rates to OCP that best reproduce states $Q_1$ and $Q_2$ ($\geq 100$ ns$^{-1}$ from a quenched compartment) are fast, consistent with highly efficient quenching. Furthermore, the measured photophysical parameters for states $Q_1$ and $Q_2$ closely match the predicted asymptotic limits for singly and doubly bound OCP + CB-PB complexes, respectively, for increasing OCP quencher strength. In this system, then, nature appears to be operating at or very near optimal conditions. The binding site symmetry that we observe here may be relevant for other related photosynthetic components; e.g., dimeric Photosystem II that shares the same $C_2$ symmetry axis as the phycobilisome. More broadly, the presence of two binding sites for OCP on the phycobilisome core may have other biological utility, for example to provide photoprotective redundancy, or to enable fine control over modulation of energy transfer to the reaction centers.

## Methods

**Strains and culture conditions**. Both CB and ΔAB strains of *Synechocystis* PCC 6803 were grown in BG11 medium at 30 °C under 30 μmol photons m$^{-2}$ s$^{-1}$,

supplemented with antibiotics as follows: 10 μg ml$^{-1}$ kanamycin (CB); 10 μg ml$^{-1}$ spectinomycin (ΔAB). CB and ΔAB were generous gifts from Dr. Ghada Ajlani. The cells were harvested at OD$_{800}$ = 1.0–1.2. The construct of the mutants used in this work has already been described: CB strain by Ughy and Ajlani[51], ΔAB strain by Ajlani et al.[68].

**Sample preparation**. Purification of PBs from the CB and ΔAB mutants was performed according to the procedure described by Ajlani et al.[68], with some modifications. Briefly, the cells were supplemented with protease inhibitor cocktail (Thermo Fisher Scientific, Waltham, MA, USA) and DNase (Sigma, St. Louis, USA), and broken in 0.8 M K-phosphate buffer at pH 7.5 by passing through three rounds of French press at 1500 psi. After 30 min incubation with 2% Triton X-100 (Sigma, St. Louis, MO, USA) at room temperature, the blue liquid supernatant recovered from 20 min centrifugation at 20,000 × *g* at 23 °C was loaded immediately onto a sucrose gradient containing layers of 1.5, 0.75, 0.5, and 0.25 M sucrose in 0.8 M potassium phosphate buffer. After ultracentrifugation (370,000 × *g*) overnight at 23 °C, the blue bands were collected from the gradient. The OCP was prepared following the procedure of Zhang et al. with minor modifications[35]. The OCP-PB complexes were prepared in 0.8 M phosphate buffer with an OCP:PB ratio of 40:1, which was previously shown to saturate OCP binding to PB[29]. This mixture was illuminated with 2000 μmol photons m$^{-2}$ s$^{-1}$ of white light for 10 min at 23 °C. Bound samples were filter-concentrated (100 kDa, Millipore Amicon Centrifugal Filters, Billerica, MA), flash-frozen and stored at −80 °C until use. All sample preparation was carried out at ~10 nM concentrations of the phycobilisome complex or greater, comparable to previous bulk studies. Immediately prior to trapping, samples were diluted in 0.8 M phosphate buffer, pH 7.5, with 1 M sucrose, to a working concentration of approximately 10 pM.

**Single-molecule trapping and data acquisition**. The ABEL trap[63,64] was implemented using a linearly polarized, 80 MHz pulsed-excitation laser (Coherent Mira OPO), loosely focused to a 0.6-μm beam waist in the sample plane. The beam was scanned in a knight's tour pattern in the trapping region of a microfluidic cell over

a 32-point $2 \times 2\,\mu m$ grid with pitch $0.4\,\mu m$. The microfluidic cell contains two perpendicular pairs of access channels that intersect at the trapping region, into which the two pairs of electrodes (Pt) for $x$- and $y$-feedback are secured. An outer ring also connects these channels to relieve pressure. The depth of the microfluidic cell in the trapping region was ~700 nm. Fluorescence photons were collected by a silicone oil immersion objective (NA 1.3, ×60, Olympus UPlanSApo), focused through a 200 μm pinhole, and separated into a spectral channel as well as two orthogonal polarization channels. Single-photon arrival (12.5 ns resolution) and delay times (4 ps resolution) were recorded in these polarization channels by two separate avalanche photodiodes (PicoQuant τ-SPAD). A custom Labview FPGA program (FPGA board National Instruments PCIe-7842R) applied feedback voltage to the $x$- and $y$-electrode pairs based on the particle position that was calculated upon the arrival of each detected photon, scanned the excitation beam, and recorded photon arrival times and delays via a commercial TCSPC system (Pico-Harp300 with PHR800 router). Fluorescence in the spectral channel was dispersed using an Amici prism (Edmund Optics NT42-586) and collected with a Si EMCCD (Andor iXon Ultra 897) at a rate of 50 Hz.

**Bulk spectrum and lifetime.** Bulk fluorescence excitation and emission spectra were taken using a Fluoromax-4 (Horiba Jobin-Yvon). Bulk lifetime data was recorded in a simple confocal configuration on the ABEL trap without feedback. Sample (~25 nM) was held in FCS sample chambers (Grace Biolabs) and measured ~20 μm above the coverslip.

**Data analysis.** For brightness determination, photon time-tags from the two polarization channels were combined into a single time-tag record and binned either into 10- or 20 ms time bins, or alternatively by arrival time into $M$-photon groups. All brightness data are reported in units of detected photon counts per second per μW of illumination (cts $s^{-1}$ $\mu W^{-1}$) to facilitate comparison across data sets. For conversion into illumination intensity, the approximate trap area was taken to be 3.36 $\mu m^2$ to reflect the grid pitch and the 32-point scan.

Polarization was calculated either by time bin or by $M$-photon group using the number of photons in the parallel channel, $N_\parallel$, and in the perpendicular channel, $N_\perp$, as:

$$\text{FPol} = \frac{N_\parallel - N_\perp}{N_\parallel + N_\perp} \qquad (1)$$

Lifetimes were fit using a minimal single exponential model, except where otherwise noted, by a maximum likelihood method after Zander et al.[71] and Brand et al.[72] as previously described[62,64,73]. Briefly, each iteration of the fit model was convolved with the experimentally measured IRF, then compared with the measured lifetime decay using a Poissonian noise model to accurately determine the likelihood of that model for single-photon counting statistics. Fitting errors are taken as the square root of each diagonal element in the inverse Fisher information matrix[74].

For the emission spectrum measurements, pixels on the EMCCD camera are mapped to wavelengths by a third-order polynomial least-squares fit to the emission pattern of calibration samples, including a HeNe cavity without mirrors and organic dyes (Alexa 633, Atto 647N) and fluorescent beads (FluoSpheres, 200 nm diameter, 625/645 nm excitation/emission). Counts in each wavelength bin were adjusted according to the derivative of this fit to account for variations in wavelength bin width. The center of mass of the emission spectrum is given by the weighted normalized sum over all wavelengths $\lambda_k$:

$$\lambda_{CM} = \frac{\sum_k \lambda_k \text{Em}(\lambda_k)}{\sum_k \text{Em}(\lambda_k)} \qquad (2)$$

The arrival time of each $M$ photon group is taken as the mean arrival time of the photons in the group. The spectral center of mass is calculated for the average measured spectrum of that group, calculated using the spectra from the camera frames in which the arrival times of the individual photons fall.

Scatter density plots are intended to convey information about the local density of data points in a scatter plot. Each point is colored according to the number of neighboring points within a neighborhood of $\pm\Delta x$ and $\pm\Delta y$, and this local density has been scaled such that 1 (yellow) is the highest density and 0 (black) is the lowest density. The size of the neighborhood affects the apparent local density, so it is important to select a neighborhood that will faithfully reproduce the ground truth local density of the underlying probability density function (PDF). We find that setting $\Delta x$ and $\Delta y$ to $0.5^*\sigma_x$ and $0.5^*\sigma_y$, respectively, closely reproduces the underlying PDF of a Gaussian function with minimal error and artifacts. However, the states measured in this study vary in their spread. Therefore, in Fig. 3, neighborhood sizes for the scatter density plots were selected to be roughly $0.5^*\sigma_{Q1}$, since it is essential in these figures to discern the approximate PDF of $Q_1$ and $Q_2$, as follows:
Br: neighborhood = ± 150 cts $s^{-1}$ $\mu W^{-1}$; $\sigma_{Br\_Q1} \approx 250$ cts $s^{-1}$ $\mu W^{-1}$ and used for $\sigma_y$. FPol: neighborhood = ± 0.025; $\sigma_{FPol\_Q1} \approx 0.05$ and used for $\sigma_x$ as appropriate. $\tau$: neighborhood = ± 0.015 ns; $\sigma_{\tau\_Q1} \approx 0.03$ ns and used for $\sigma_x$ as appropriate. $\lambda_{CM}$: neighborhood = ± 0.5 nm; $\sigma_{CM\_Q1} \approx 1$ nm and used for $\sigma_x$ as appropriate.
To determine the expected photophysical parameters for the observed states, histograms in Br, $\tau$, $\lambda_{CM}$, and FPol were formed from 200-photon grouped data (Supplementary Figures 5–8) for each state: U, B, $Q_1$, and $Q_2$. A Gaussian was fit in

each dimension to determine the most probable parameter values and standard deviations, as reported in Table 1.

**Simulations.** The compartmental model for the phycobilisome was based directly on an existing compartmental model of the phycobilisome developed by van Stokkum et al.[61]. The connectivity (Fig. 5a) and transfer rates between compartments are detailed in Supplementary Table 1. Simulations using this model consisted of three steps. First, a single compartment was populated with an excitation, simulating an absorption event. The probability a compartment was populated was determined by the estimated molar extinction coefficient at 590 nm for that compartment normalized by the total CB-PB molar extinction coefficient. These probabilities are listed in Supplementary Table 2. Second, the excitation underwent a random walk with 100 fs time steps where the probability of transferring to an adjacent compartment or fluorescing (with 50% quantum yield) was determined by the coupling strengths shown in Supplementary Table 1. Back transfer was not permitted from OCP compartments. For random walks terminating with the emission of a fluorescent photon, the emission compartment and time were recorded, along with the delay time between the initial absorption event and emission, which was used for the calculation of lifetime. $10^5$ absorption events were simulated for each condition tested. By normalizing the brightness of the simulated unquenched state to the experimentally measured unquenched state, we were able to directly compare the predicted brightness for each of the other three states (B, $Q_1$, and $Q_2$) to their experimentally measured brightnesses. With this scaling, the simulated data could be passed through the same data analysis pipeline described above to generate the scatter plot data shown in Fig. 5b.

**Reporting summary.** Further information on experimental design is available in the Nature Research Reporting Summary linked to this article.

## Data availability
Data supporting the findings of this manuscript are available from the corresponding author upon reasonable request. A reporting summary for this Article is available as a Supplementary Information file. The source data underlying Figs. 1de, 2bc, 3ade, 4abc, 5bcd and Supplementary Figures are provided as a Source Data file.

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

## Acknowledgements

We thank Ghada Ajlani for the CB and ΔAB Synechocystis mutants. This material is based upon work supported in part by the U.S. Department of Energy, Office of Science, Office of Basic Energy Sciences, Chemical Sciences, Geosciences, & Biosciences Division under Award Number DE-FG02-07ER15892 (Physical Biosciences Program, WEM) and Award Number DE-FG02-07ER15902 (Photosynthetic Systems Program, REB and HL).

## Author contributions

A.H.S., P.D.D., R.E.B. and W.E.M. designed the research, A.H.S. performed experiments and analyzed the data, and drafted the manuscript, P.D.D. and A.H.S. implemented the compartmental model, A.H.S. and P.D.D. made figures, H.L. and N.M. cultured cells, purified proteins, and prepared quenched phycobilisome samples. All authors discussed and interpreted results, and reviewed and edited the manuscript.

## Additional information

**Competing interests:** The authors declare no competing interests.

