## [Peer Review File · Nature Communications]

Reviewers' Comments:

Reviewer #1:

Remarks to the Author:

This manuscript addresses an important question in the field of photoprotection in cyanobacteria, the binding site to the phycobilisomes (PBS) and the quenching mechanism of the Orange Carotenoid Protein (OCP). The work applies a cutting-edge technique, single-molecule trapping, and spectroscopy, to obtain new insights about where OCP binds in the phycobilisomes. The main finding of the manuscript is the observation of two distinct binding sites for the OCP on the phycobilisomes. This work is novel in its approach and conclusions. I have a few comments for the Authors' consideration:

General comments

- The main finding is that two OCP binds to the PBS, therefore, therefore two distinct binding sites on the PBS are required. This place is most likely in the core. Unfortunately, the exactly structural place in the PBS is still unknown. Can the authors' previous work on the structure of the OCP and its crosslinking with the PBS, as well as the PBS structure be more fully discussed in the context of these findings?
- There are reports of different oligomeric states of the OCP and that the N-terminal half of the protein in isolation also quenches. These data should be considered in the context of the Discussion. Moreover, one assumes that when the authors describe OCP binds to the PBS, they are referring to the red form. The form they are referring to should be clarified in the text.
- The fluorescence lifetime of Q1 and Q2 are different, Q1 is longer than Q2 (0.21 ns). Does it mean that Q1 is a predominant state? If so, this will imply that the two OCP bound to the PBS is not the main scenario for quenching. Could it be that these two Q1 and Q2 represents a turnover of OCP in the PBS quenching mechanism?
- This new scenario opens questions related to FRP (Fluorescence Recovery Protein). If there are two OCPs attached to the PBS, does the system need two FRP, or one FRP is able to bind to two OCPs? This might also be considered a constraint in picturing how the OCP binds to the PBS.

Minor comments

Abstract

- Sentence 'We observed not one but two distinct OCP-quenched states', consider remove 'not one but' Or clarify why this is a "but" statement
- One of the OCP-quenched states has a lifetime of 0.09 ns and 6% of unquenched brightness, in conclusion section says 5.7%, consider homogenize this value.

Introduction

- 'and actuated by a single water-soluble photoactive protein, known as the Orange Carotenoid Protein (OCP)', add reference 21.
- 'entire class of primary producers in many ecosystems', references 12-14 do not seem appropriate. There have been at least 3 reviews in the last two years (2016-2018) focusing on ecophysiological diverse cyanobacteria and the diversity of the OCP, including new families.
- In the figure 1a, each domain named with the acronym NTD and CTD, consider clarify these terms in the second paragraph in the introduction and in the figure legend.
- 'non-covalently binds a keto carotenoid, 3' hydroxy-echinenone (3'- hECN), consider add reference 21. OCP has been shown to bind various keto-carotenoids when purified from cyanobacteria (hECN, ECN, CAN) the references: 21 and Punginelli et al., 2009.
- '1) Can OCP bind to multiple structurally and chemically unique sites on the PBs, or is there a single binding site?', consider remove 'unique'.
- Can the author clarify the terms of 'potential heterogeneity'?
- Introduction, paragraph 4: the author should first describe the structure of intact PB (both core and rods) before introducing the truncated PB (CB-PB). The author should explain the structural difference between wt-PB and CB-PB and the reason for using the CB-PB in this study instead of wt-PB. Line 3 and 4: change the words "rods" to "cylinder" to distinguish from phycocyanin rods of

PB.

- Why use of PBS mutant for structural consistency? It seems that the CB-PB mutant lacks the rods, therefore, that means that the authors assume that the binding is going to occur in the core of the PBS and the other parts from the PBS have no contribution in the quenching.

Results

- It is not clear how OCP was photoactivated to perform the experiments.

Discussion

First paragraph: With the information presented in this paragraph, can the author conclude that the rods are not involved in the PBS mechanism quenching?

Fifth paragraph: 'It is also potentially ...or through charge transfer', this sentence lacks references.

Discussion, last paragraph: Could the author comments on which of the three combinations (ad, a'd', and bc) is the most likely quenching sites based on the simulation results shown in Fig. 5d.

Conclusion, paragraph 2: the timescale "1 - 104 ms" was mentioned here for the first time without giving any details in the previous sections.

Materials and methods

- Sample preparation:

o Replace 0.5 h for 30 min

o g in italics

o Why the addition of 1 M sucrose?

- The details how the OCP was photoactivated to get the red form of OCP are missing.

Reviewer #2:

Remarks to the Author:

This is a very interesting study on the quenching of excitation energy in phycobilisomes by the orange carotenoid protein (OCP).

Using single molecule trapping and spectroscopy two subpopulations of quenched complexes (and one of an unquenched complex) are identified which exhibit different brightness and excited state lifetime. The relative contribution of these subpopulations is found to change with time (from 10 min to 300 min) and this change is explained by assuming that there are two distinct binding sites for OCP which, if both are occupied, give the strongest quenching. With increasing time one OCP detaches and the excited state lifetime of phycobilisome increases accordingly. This model is quantitatively confirmed by performing kinetic Monte Carlo simulations of excitation energy transfer and quenching using an existing compartment model for the different states from the literature. With increasing time some of the phycobilisome rods are assumed to detach from the phycobilisome core and give rise to the third subpopulation detected with long excited state lifetime. The blue shifted fluorescence spectrum of the latter population is a nice confirmation of this hypothesis.

The paper is very well written, the results are discussed critically and the proposed model seems to be the most simple and logical that can explain the spectra.

The results are significant because they provide new microscopic insight into the quenching of phycobilisomes, which is an important regulation mechanism of photosynthesis.

The manuscript can be published. Two minor points that could be helpful in improving the manuscript even further are

(i) The fluorescence decay of the bulk sample (Fig. 1e) characterized as "fully quenched" is bi-exponential with a large contribution from a long-lived (1.45 ns) component that is not found in the single molecule spectra of the quenched complexes. As the authors state, this long lifetime is similar to the lifetime of unquenched complexes. However, it is also estimated that the fluorescence of the "fully quenched" sample is only 9 percent of the unquenched sample. So, it

would be too simple to assume that in the "fully quenched" sample there is a large subpopulation of unquenched complexes that give rise to the 1.45 ns lifetime component in the fluorescence. So how can the latter be understood?

(ii) Can the authors exclude from their kinetic Monte Carlo analysis that there are more than two bound OCPs?

Reviewer #3:

The manuscript by Squires and colleagues presents a single molecule spectroscopy study of the interaction between two isolated physiological players: OCP and PBS, which are involved in photoprotection in cyanobacterial cells. In its active form, OCP binds to a PBS and induces energy dissipation that is signified by the PBS' fluorescence quenching. Single molecule fluorescence spectroscopy experiments are performed in an ABEL trap, a system that allows tracking the fluorescing complexes in solution and record fluorescence brightness, fluorescence polarization, fluorescence lifetimes and fluorescence spectra. Here, the SMS allowed the authors to investigate some mechanistic details of the interaction between OCP and PBS. The discussion of the SMS results leads the authors to the conclusion that under saturating conditions more than one OCP molecule binds to a core of PBS. While this idea is, as such, interesting and, given the symmetry of the PBS' cores, not impossible, the results and analysis presented in the manuscript do not convincingly support this hypothesis. Moreover, some results seem to argue against the proposed model of 2 OCPs binding to the PBS. Finally, the manuscript does not present sufficient controls allowing the readers/ reviewers to assess the impact of the ABEL-trap condition on the interaction between OCP and PBS.

Previous in vitro characterization of the OCP-PBS system have shown that upon re-isolation of the OCP-PBS complexes from the PBS exposed to an excess of OCP, each PBS bound, on average, 1.2 OCPs. While 1.2 is clearly higher than 1 suggesting that some PBS complexes could have attached more than 1 OCP (whether it is a specific binding or not, remains unclear), the spectroscopic properties of OCP-PBS complexes were always identical, also in two scenarios: 1) just after formation of a quenched complex under saturating conditions, and 2) after re-isolation, which takes a few hours. Following the interpretation from the submitted manuscript, after the re-isolation, the OCP-PBS complexes should be in state Q1, as shown by the OCP:PBS ratio of 1.2:1. However, just after the formation of quenched OCP-PBS complexes, they are expected to be in a spectroscopically different state Q2 (binding more OCPs), which is apparently not observed in ensemble fluorescence spectra published previously. It is unclear how a difference between Q1 and Q2 would be visible in SMS and not in ensemble. Moreover, as can be concluded from the fluorescence lifetimes (Tian et al 2011 and 2012) and following kinetic modelling (van Stokkum et al 2018) of OCP-related quenching, the presence of a second OCP in the PBS core would not change the efficiency of quenching, i.e. in ensemble experiments, whether one or two OCPs bind to the core, the outcome would be indistinguishable.

An insufficient characterization of the unquenched PBS complexes in the manuscript does not allow the reader to critically assess the results for the OCP-PBS. In Fig 2b, the authors suggest that PBS alone can only assume one photophysical state, namely the unquenched state (U). This result is inconsistent with the previous SMS reports for these complexes where excited PBS were reversibly switching between an unquenched and a quenched state (Gwizdala et al 2016 and 2018), with the switching rates increasing with the excitation intensity. In the example from Fig 2b no such switching or other-than-U states were shown. For a reason not explained in the manuscript, the experiments on the OCP-PBS complexes were performed using 3 times higher light intensity than for PBS alone, i.e., under conditions, in which the probability of finding PBS in the quenched states is higher. It is therefore possible that the Q1 state is related to light-induced dynamics of PBS and not to the activity of OCP. Moreover, the duration of different states presented in Fig 2c supports the idea that some of these states, in particular Q1, are indeed light-induced and that OCP is not present in these complexes (typically OCP forms very stable complexes with PBS). Following this thought, can it be excluded that only Q2 states would be OCP-related and Q1 states would represent intrinsic dynamics of PBS (induced by higher

light intensity than in the control from Fig 2b). Moreover, the relative difference in fluorescence lifetimes between Q1 and Q2 (Q1 having ~2 times longer lifetime than Q2) is in agreement with the relative difference between the lifetimes of OCP-induced and light-induced (intrinsic, measured in absence of OCP) quenched states of PBS reported previously (Gwizdala et al 2018).

The authors described that they did not observe a transition between Q1 and Q2 state. However, if 2 OCPs were to bind to a PBS it is far more likely that they would bind sequentially and not at the same time, i.e., first one OCP would bind and then the second – and the same for detachment. Since no such observation was reported it has to be assumed that the states Q1 and Q2 are accessed due to single events: for Q1, light-induced change of properties of a pigment in PBS; for Q2, binding of OCP. Assuming the two OCP scenario as proposed by the authors, transitions between the Q1 and Q2 states would be frequent due to the sequential attachment of OCPs to the PBS.

Finally, the manuscript does not test the proposed model against its biological relevance. In the cells the amount of OCP is smaller than that of PBS. This fact is included in the models of Tian et al. 2011 showing that even at low concentrations in the cells, OCP creates an efficient channel for energy dissipation from PBS, in effect providing photoprotection. If however, the number of PBS that bind OCP decreases by half (since every PBS can bind 2 OCPs), it is doubtful that OCP would still serve a photoprotective function in the cells.

As summarized above the manuscript does not convincingly proof the existence of two quenched states induced by one or two OCPs. Instead, it may show the difference between light-induced and OCP-induced dynamics, a topic previously addressed in SMS studies of the interaction between OCP and PBS. On top of that, the manuscript shows a few minor weaknesses that should be fixed:

- 1) The manuscript does not address the possibility that the electric field present in the ABEL-trap could impact the: 1) photoactivation of OCP, 2) binding of OCP to PBS, 3) electronic interaction between the OCP and PBS, and 4) the physicochemical properties of PBS. Relevant controls are missing.
- 2) Regarding the characterization of the PBS in the absence of OCP, the authors did not explain why detached C-PC rods are more abundant in quenched complexes than in PBS alone. Since OCP does not directly interact with C-PC, this suggests that the conditions in OCP-related experiments had a larger impact on the PBS.
- 3) Description of methods is not sufficient. E.g. it is not explained how, when and under which conditions, the photoactivation of OCP was induced.
- 4) SMS figures show polarization measurements that are not discussed in the text.
- 5) It is not sufficiently explained why the authors used the “CB-PB” complexes instead of PBS isolated from a WT.
- 6) I would recommend not using the center of mass approach to analyze the spectra that are as asymmetrical as the spectra of PBS.
- 7) In the introduction, when describing the core of PBS the “cylinders” are referred to as “rods”, which is highly confusing, given that it is C-PC and not APC that forms rods. In the discussion the cylinders are referred to as cylinders.

Response to reviewers

Identification of two distinct binding sites of Orange Carotenoid Protein on the phycobilisome by single-molecular trapping and spectroscopy

Reviewer #1 (Remarks to the Author):

This manuscript addresses an important question in the field of photoprotection in cyanobacteria, the binding site to the phycobilisomes (PBS) and the quenching mechanism of the Orange Carotenoid Protein (OCP). The work applies a cutting-edge technique, single-molecule trapping, and spectroscopy, to obtain new insights about where OCP binds in the phycobilisomes. The main finding of the manuscript is the observation of two distinct binding sites for the OCP on the phycobilisomes. This work is novel in its approach and conclusions. I have a few comments for the Authors' consideration:

We very much appreciate Reviewer #1's careful reading of the manuscript, helpful suggestions and questions. We have addressed their questions and comments below. Yellow highlights indicate responses that correspond to changes in the manuscript or SI.

General comments

- The main finding is that two OCP binds to the PBS, therefore, therefore two distinct binding sites on the PBS are required. This place is most likely in the core. Unfortunately, the exactly structural place in the PBS is still unknown. Can the authors' previous work on the structure of the OCP and its crosslinking with the PBS, as well as the PBS structure be more fully discussed in the context of these findings?

It's true that the exact structure of PBS in cyanobacteria is still unknown. Although the 3D structure of PBS from a red alga (*Griffithsia pacifica*) has been reported (Zhang et al., 2017, *Nature*), OCP is absent in red alga.

The studies of Blankenship and co-workers revealed a close association of ApcB and the flexible linker region of OCP between the N- and C-terminal domains, enabling these authors to propose that "there are only two OCP binding and, thus, quenching sites per PBS" (Zhang et al. 2014). Mass spectrometry in combination with other biophysical techniques has revealed dramatic domain rearrangements of OCP upon photoactivation, enabling OCP^R binding to PBS (Liu, et al., 2014, BBA; Gupta et al., 2015 PNAS). Similar results highlighting the importance of ApcB and the linker region of OCP have also been reported using different cross-linkers (Harris et al., 2016 PNAS) and multiple OCP-PBS binding models have been proposed that involve quenching of Apc660 or Apc680 chromophores on the basal cylinders, with slight variations on the precise position of the OCP (see, for example, Zhang et al 2014 Nature, Harris et al 2016 PNAS, Kuzminov et al 2012 BBA). While the results of the current study cannot directly distinguish among these, the binding states we observe and our computational results are directly consistent with these models. We have briefly highlighted this point in our discussion to clarify this:

Discussion, p. 6

This is consistent with previous suggestions that OCP binds the basal rods³⁸, that it might quench either Apc₆₆₀^{31,33} and/or Apc₆₈₀³⁹, and that it may bind near ApcE^{35,36}.

- There are reports of different oligomeric states of the OCP and that the N-terminal half of the protein in isolation also quenches. These data should be considered in the context of the Discussion.

This is a useful point related to the possibility already considered in the paper that OCP might have variable quenching strengths. Could different oligomerization states of OCP be responsible for the different quenched states? We believe that the answer is no, because presumably an OCP oligomer could only quench at one spatial location, and our modeling results already show that no matter how strong the quencher is, a quencher at only one location is insufficient to produce state Q2. Therefore, although an OCP oligomer might have different quenching properties from a monomer, this could not explain state Q2. We have added a few sentences to clarify this interesting point in our Discussion.

Discussion, p. 6

... This might be possible if, for example, the rate of energy transfer toward the site of quenching were significantly increased in Q₂ as compared to Q₁. A related possibility is that OCP itself can exhibit variations (for example, by oligomerization, conformational changes, or changes in electronic structure) that create precisely two different quenching efficiencies. To test these hypotheses, we simulated the effect of a single bound OCP with variable quenching strength, as described below.

Discussion, p. 7

... First, we asked whether or not Q₁ and Q₂ could result from a single bound quencher located at different quenching sites, or with different quenching strength due to oligomerization or structural variations...

Moreover, one assumes that when the authors describe OCP binds to the PBS, they are referring to the red form. The form they are referring to should be clarified in the text.

Yes, in all bound samples, the OCP is activated (red form). We have added language to the introduction to indicate that "OCP" always refers to the active form following the first discussion of its activation, except where otherwise specifically noted.

Introduction, p. 1:

Activated OCP^R (referred to hereafter as simply OCP; the inactive form will be specifically annotated as OCP^O) can bind to the phycobilisome (PB), the primary light-harvesting antenna of cyanobacteria, and quenches the excitation energy of the complex (Fig. 1b).

- The fluorescence lifetime of Q1 and Q2 are different, Q1 is longer than Q2 (0.21 ns). Does it mean that Q1 is a predominant state? If so, this will imply that the two OCP bound to the PBS is not the main scenario for quenching. Could it be that these two Q1 and Q2 represents a turnover of OCP in the PBS quenching mechanism?

To clarify, the Q2 quenching process means that any excitons generated can be quenched by either one of the two OCPs bound, resulting in a shorter fluorescence lifetime than Q1, the case when only one OCP is bound. Thus, the relative magnitudes of fluorescence lifetimes are unrelated to the proportional populations of complexes producing those lifetimes. As shown by our data (Fig. 4) most complexes are initially in Q2 under saturating binding conditions.

We have added language to the abstract to clarify this:

Abstract, p. 1

Photon-by-photon Monte Carlo simulations of exciton transfer through the phycobilisome suggest that the observed quenched states are kinetically consistent with either two or one bound OCPs, respectively, underscoring an unprecedented mechanism for excitation control in this key photosynthetic unit.

- This new scenario opens questions related to FRP (Fluorescence Recovery Protein). If there are two

OCPs attached to the PBS, does the system need two FRP, or one FRP is able to bind to two OCPs? This might also be considered a constraint in picturing how the OCP binds to the PBS.

We agree that the question of how FRP interacts with OCP is potentially quite interesting, and we look forward to possible future studies that investigate that topic.

Minor comments

Abstract

- Sentence 'We observed not one but two distinct OCP-quenched states', consider remove 'not one but' Or clarify why this is a "but" statement

We have changed this sentence to read (noting the 150 word limit in the Abstract):

Abstract, p1:

Surprisingly, we observed two distinct OCP-quenched states, with lifetimes 0.09 ns (6% of unquenched brightness) and 0.21 ns (11% brightness).

- One of the OCP-quenched states has a lifetime of 0.09 ns and 6% of unquenched brightness, in conclusion section says 5.7%, consider homogenize this value.

We thank the reviewer for noting this inconsistency. We have homogenized these values throughout the paper to use the 6% value, which best represents our uncertainty.

Introduction

- 'and actuated by a single water-soluble photoactive protein, known as the Orange Carotenoid Protein (OCP)', add reference 21.

We agree that Reference 21 is appropriate here, and have added it here in the manuscript. It is now renumbered as Reference #12.

Introduction, p1:

Unlike higher plants, cyanobacteria exhibit a novel NPQ mechanism⁷⁻⁹ that is triggered, transduced, and actuated by a single water-soluble photoactive protein, known as the Orange Carotenoid Protein (OCP)^{8,10-12}.

- 'entire class of primary producers in many ecosystems', references 12-14 do not seem appropriate. There have been at least 3 reviews in the last two years (2016-2018) focusing on ecophysiological diversity of cyanobacteria and the diversity of the OCP, including new families.

The first three of these references (now #13-15) establish cyanobacteria as primary producers in certain marine ecosystems by estimating their prevalence and respiratory contributions to carbon fixation, supporting the idea that cyanobacteria are a dominant photosynthetic species in these ecosystems. The last reference (#16) is representative of the recent publications mentioned here that describe the broad ecophysiological variety of cyanobacterial species. Taken together, we feel that these references adequately support the idea that cyanobacteria are an "entire class of primary producers" in "many ecosystems".

- In the figure 1a, each domain named with the acronym NTD and CTD, consider clarify these terms in the second paragraph in the introduction and in the figure legend.

We thank the reviewer for catching this omission. We have added clarification of these acronyms to the text of the introduction and the figure legend.

Introduction, p.1

Cytosolic OCP in its inactive orange form (OCP⁰) non-covalently binds a keto carotenoid, 3' hydroxy-echinenone (3'-hECN^{7,8}), that bridges the α -helical N-terminal domain (NTD) and the α/β C-terminal domain (CTD),...

Figure 1 Caption

...A keto-carotenoid spans the interior of the N- and C-terminal domains (NTD and CTD, respectively),...

- 'non-covalently binds a keto carotenoid, 3' hydroxy-echinenone (3'- hECN), consider add reference 21. OCP has been shown to bind various keto-carotenoids when purified from cyanobacteria (hECN, ECN, CAN) the references: 21 and Punginelli et al., 2009.

We have added reference 21 (renumbered: #12) here as suggested. As we do not discuss binding of alternative keto-carotenoids in this work, and due to limitations on the total number of permitted references, we have not added Punginelli et al. (2009).

Introduction, p.1

Cytosolic OCP in its inactive orange form (OCP⁰) non-covalently binds a keto carotenoid, 3' hydroxy-echinenone (3'-hECN^{12,20,21}),

- '1) Can OCP bind to multiple structurally and chemically unique sites on the PBs, or is there a single binding site?, consider remove 'unique'.

Upon consideration, we have altered this word to "distinct", which may more clearly convey the point of the question posed: It is already known that OCP binds, but it is not known where. Therefore, this question asks whether it can bind at structurally and chemically distinct sites on the core.

Introduction, p.2

1) Can OCP bind to multiple structurally and chemically distinct sites on the PBs, or is there a single binding site?

- Can the author clarify the terms of 'potential heterogeneity'?

We have added language to the introduction to make this sentence more specific, as follows:

Introduction, p.2

The common thread in all three questions is potential structural, conformational, or binding heterogeneity in this system, which if present might be a source of the divergent results reported in the literature.

- Introduction, paragraph 4: the author should first describe the structure of intact PB (both core and rods) before introducing the truncated PB (CB-PB). The author should explain the structural difference between wt-PB and CB-PB and the reason for using the CB-PB in this study instead of wt-PB. Line 3 and 4: change the words "rods" to "cylinder" to distinguish from phyocyanin rods of PB. Why use of PBS

mutant for structural consistency? It seems that the CB-PB mutant lacks the rods, therefore, that means that the authors assume that the binding is going to occur in the core of the PBS and the other parts from the PBS have no contribution in the quenching.

We have altered the text of the introduction to first describe the WT structure, as suggested, and have changed the term “rods” to “cylinders” when describing APC. As stated in the introduction, the CB-PB mutant was employed for structural homogeneity; we have added additional language to clarify this. Regarding binding in the core, yes, multiple previous studies have clearly indicated that OCP binds the core of the phycobilisome, as referenced in our Introduction (p.1).

Introduction, p.2

Excitons are funneled into the PB core by six C-phycoyanin rods⁹, which in the wild-type *Synechocystis* PCC 6803 each consist of three stacked C-phycoyanin hexamers. Here, we employ the “truncated” CB-PB mutant of *Synechocystis* PCC 6803⁵¹ to ensure structural homogeneity (see Methods for details). This is necessary because it has been previously suggested that both the architecture and the spectroscopic properties of phycobilisomes, especially the C-phycoyanin rods, may be dynamic and/or modular^{47,52-54}, facilitating structural remodeling *in vivo*^{54,55}.

Introduction, p.2

In *Synechocystis* PCC 6803, the PB core comprises three cylinders that each contain four allophycocyanin (Apc) disc-shaped $\alpha\beta$ trimers, most of which emit at 660 nm. The two discs at counterparallel ends of each of the two bottom cylinders together contain ...

Results

- It is not clear how OCP was photoactivated to perform the experiments.

As described in our Methods section, OCP was isolated, photoactivated, and bound to phycobilisomes according to the same protocol described in Reference 35 (Zhang et al., Biochemistry 2013). For clarity, we have added language to the methods section to more explicitly describe this process:

Methods, p. 9

The OCP was prepared following the procedure of Zhang et al. with minor modifications³⁵. The OCP-PB complexes were prepared in 0.8 M phosphate buffer with an OCP:PB ratio of 40:1, which was previously shown to saturate OCP binding to PB²⁹. This mixture was illuminated with 2000 $\mu\text{mol photons m}^{-2}\cdot\text{s}^{-1}$ of white light for 10 min at 23 °C. Bound samples were filter-concentrated (100 kDa, Millipore Amicon Centrifugal Filters, Billerica, MA), flash-frozen and stored at -80 °C until use.

Discussion

First paragraph: With the information presented in this paragraph, can the author conclude that the rods are not involved in the PBS mechanism quenching?

This is an interesting question - our data indicates that the rods are unlikely to be the site of quenching, and that energy likely transfers to the OCP from the core rather than from rods, but we cannot rule out the possibility that the rods do play some role in quenching – for example, they might be structurally involved in the binding process, or some small amount of energy may transfer to the OCP from proximal rods. As our measurements are exclusively photophysical, we do not have detailed information on the exact structural arrangement. We therefore refrain in this work from speculation on the role or lack thereof that the rods may play in quenching.

Fifth paragraph: ‘It is also potentiallyor through charge transfer’, this sentence lacks references.

This was an oversight - we have added the appropriate references to this as follows:

Discussion, p. 6

...that hECN is involved in quenching one or more proximal chromophores upon binding, either through resonant or directly coupled energy transfer^{30,69,70}, or through charge transfer^{31,33}.

Discussion, last paragraph: Could the author comments on which of the three combinations (ad, a'd', and bc) is the most likely quenching sites based on the simulation results shown in Fig. 5d.

We agree that this is an interesting question. All three of these combinations produce modeling predictions that are consistent with the observed data. We therefore do not speculate here as to which combination is most likely. Furthermore, the model used in this study is compartmentalized and therefore coarse-grained compared to the actual energy transfer processes in the phycobilisome: in reality, it is likely that one or a few pigments are the primary sites of energy transfer to the OCP, which cannot be captured by this model. Rather, from these results we can conclude that binding at any pair of antiparallel ends of the bottom two cylinders in the core could produce sufficient quenching to explain state Q2.

To clarify, we have added this language at the end of the Discussion:

Discussion, p. 7

These results imply that binding at any pair of antiparallel ends of the bottom two cylinders in the core could produce sufficient quenching to explain state Q2. This is consistent with previous suggestions that OCP binds the basal rods³⁸, that it might quench either Apc₆₆₀^{31,33} and/or Apc₆₈₀³⁹, and that it may bind near ApcE^{35,36}.

Conclusion, paragraph 2: the timescale "1 - 104 ms" was mentioned here for the first time without giving any details in the previous sections.

We have modified this sentence to clarify that 1-10⁴ ms (0.001-10 seconds) refers to the timescales relevant to the ABEL trap data reported in this paper.

Conclusion, p. 8

These data also show that if multiple bound conformations of OCP exist, as has been suggested in particular for the C-terminal domain, these conformations do not produce spectroscopically distinct quenched states on timescales relevant to the measurements described here (between 1 - 10⁴ ms).

Materials and methods

- Sample preparation:

o Replace 0.5 h for 30 min

Fixed.

o g in italics

Fixed.

o Why the addition of 1 M sucrose?

Sucrose and other sugars are well-established as stabilizing agents for proteins and protein complexes (Arakawa and Timasheff 1982, *Biochemistry*) that achieve their effect by slightly altering the hydration of the protein surface. We observe that 1 M sucrose produces long-lived OCP-PB binding under our dilute conditions.

- *The details how the OCP was photoactivated to get the red form of OCP are missing.*

We have added these details to the Methods section, as detailed above.

Reviewer #2 (Remarks to the Author):

This is a very interesting study on the quenching of excitation energy in phycobilisomes by the orange carotenoid protein (OCP).

Using single molecule trapping and spectroscopy two subpopulations of quenched complexes (and one of an unquenched complex) are identified which exhibit different brightness and excited state lifetime. The relative contribution of these subpopulations is found to change with time (from 10 min to 300 min) and this change is explained by assuming that there are two distinct binding sites for OCP which, if both are occupied, give the strongest quenching. With increasing time one OCP detaches and the excited state lifetime of phycobilisome increases accordingly. This model is quantitatively confirmed by performing kinetic Monte Carlo simulations of excitation energy transfer and quenching using an existing compartment model for the different states from the literature. With increasing time some of the phycobilisome rods are assumed to detach from the phycobilisome core and give rise to the third subpopulation detected with long excited state lifetime. The blue shifted fluorescence spectrum of the latter population is a nice confirmation of this hypothesis.

The paper is very well written, the results are discussed critically and the proposed model seems to be the most simple and logical that can explain the spectra.

The results are significant because they provide new microscopic insight into the quenching of phycobilisomes, which is an important regulation mechanism of photosynthesis.

The manuscript can be published.

We very much appreciate Reviewer #2's careful reading of the manuscript, suggestions to improve the work, and evaluation of the work's significance and novelty. We have addressed their suggestions and questions below. Yellow highlights indicate responses that correspond to changes in the manuscript or SI.

Two minor points that could be helpful in improving the manuscript even further are

(i) The fluorescence decay of the bulk sample (Fig. 1e) characterized as "fully quenched" is bi-exponential with a large contribution from a long-lived (1.45 ns) component that is not found in the single molecule spectra of the quenched complexes. As the authors state, this long lifetime is similar to the lifetime of unquenched complexes. However, it is also estimated that the fluorescence of the "fully quenched" sample is only 9 percent of the unquenched sample. So, it would be too simple to assume that in the "fully quenched" sample there is a large subpopulation of unquenched complexes that give rise to the 1.45 ns lifetime component in the fluorescence. So how can the latter be understood?

We thank the reviewer for raising this question, as it led us to further cross-check our bulk and single-molecule measurements in a way that highlights their mutual consistency. Only a very low percentage of unquenched complexes are necessary to produce the 1.45 ns component of the decay, because the unquenched complexes are much brighter than the quenched complexes (note also that the fluorescence decay curves are shown in log scale, so the area under the long-lifetime component is somewhat magnified in this view). We have added the following Note to our Supplementary Information to quantitatively assess the consistency of our bulk data, and we cite this SI Note on p. 3

SI Note 1: Bulk measurements show a low percentage of unquenched complexes in the quenched samples

The bulk measurements made in this study (lifetime, emission spectrum) are mutually consistent with a very low percentage of unquenched complexes initially present in the quenched samples

(~3.7%). The fluorescence decay of the PB+OCP bulk sample was fit with a bi-exponential decay, which prior to convolution with the IRF and addition of the expected background can be simply written as:

$$P(t) = Ae^{-t/\tau_1} + (1 - A)e^{-t/\tau_2}$$

The fitted parameters for this sample were $A = 0.9603$, $\tau_1 = 0.096$ ns, and $\tau_2 = 1.45$ ns. By comparing the contributions to the total area under the curve from each lifetime component, we can calculate the proportion of photons under this curve that come from the short lifetime, which we assume to be a quenched state:

$$\% \text{ photons from state Q} = \frac{A\tau_1}{A\tau_1 + (1 - A)\tau_2} \times 100\% = 61.5\% \text{ of photons}$$

...and similarly, from the long lifetime, which we assume to be the unquenched state:

$$\% \text{ photons from state U} = \frac{(1 - A)\tau_2}{A\tau_1 + (1 - A)\tau_2} \times 100\% = 38.5\% \text{ of photons}$$

However, to find the relative number of complexes of each type, we must scale by the relative brightness of each state. The fully quenched complex is 6% of the brightness of an unquenched complex. Therefore:

$$\% \text{ state Q complexes} = \frac{A\tau_1/0.06}{A\tau_1/0.06 + (1 - A)\tau_2} \times 100\% = 96.3\% \text{ of complexes}$$

and

$$\% \text{ state U complexes} = \frac{(1 - A)\tau_2}{A\tau_1/0.06 + (1 - A)\tau_2} \times 100\% = 3.7\% \text{ of complexes}$$

That is, this bulk lifetime decay shows that <5% of complexes are unquenched. We can then calculate the expected total brightness ratio of this quenched sample to an unquenched sample, again assuming that quenched complexes are 6% as bright as unquenched complexes:

$$96.3\% * 0.06 + 3.7\% * 1 = 9.5\%$$

We expect that the bulk sample from the lifetime decay shown in Figure 1e will be about 9.5% as bright as the unquenched sample. This is very close to the ratio of the integrated areas under the spectra shown in Figure 1d, which shows that the quenched sample is 9% as bright as the unquenched sample.

Additional complexities including the presence of some Q_1 complexes and/or the presence of decoupled rods are not included here, and indeed might alter these numbers a little. Nevertheless, the close agreement of these calculations to the measurements serves to demonstrate that the bulk fluorescence lifetime data and the bulk fluorescence emission spectrum data are mutually consistent, and indicate that very few unquenched complexes are present in the quenched sample.

Moreover, these numbers also imply that the bulk data are consistent with the reported single-molecule quenching levels (6% for Q_2), as well as with our suggestion that most complexes are quenched into the Q_2 state upon (saturated ratio) binding of OCP.

(ii) Can the authors exclude from their kinetic Monte Carlo analysis that there are more than two bound OCPs?

This is another great question, which we have done additional simulations to test. When we test scenarios that involve more than two quenching sites, or which involve two asymmetric sites, we predict that more than two quenched populations should be observed in the single-molecule data.

For more than two sites (for example, for three bound) you would expect to see 1) three bound OCPs, 2) at least one (but likely more than one) quenching level that represents two bound OCPs and 3) at least one

quenching level that represents one bound OCP (again, likely more than one). So, for three bound OCPs, we would expect to see at least three or more distinct populations. While this possibility cannot be completely eliminated (because it is conceivable that strong cooperative binding or similar anomalous factors might prevent some of these combinatorically possible states from being populated), in our view the simplest possible explanation for seeing precisely two populations is that two OCPs can bind at symmetric sites.

To demonstrate this, we ran a three-quencher scenario that includes two symmetric sites (therefore producing some degeneracy and lowering the number of expected states). We selected the quencher strength by requiring that the triply-quenched complex produce the Q2 photophysical parameters, and then used this quencher strength to simulate the combinations of two and one quenchers. As described above, the results predict that this scenario would produce at least 4 types of quenching populations, which we do not observe in the data.

We have added an additional figure to the SI to display this simulation result (Figure S11). We have also added language to the Discussion section of the main manuscript regarding this point, along with an SI Note with additional discussion.

Discussion, p. 8

...The other three combinations (ad , $a'd'$, and bc) all asymptotically approach spectroscopic parameters that are very similar to Q₂ at high quenching strength. Here, only rotationally symmetric sites were considered because this degeneracy produces precisely two observable states (two or one bound OCP), while additional or non-degenerate sites would most likely produce more than two quenched states, as discussed in more detail in SI Note 4. For the purposes of the results shown in Fig. 5b, we have taken the effective quenching rate...

SI Figure S11

Figure S11: Compartmental model of OCP-quenched CB-PB phycobilisome for three quenchers at a' , d' , and e . a) Brightness and fluorescence lifetime of the CB-PB phycobilisome for three bound quenchers located at sites $a'd'e$, plotted for increasing quencher strength. This panel illustrates that a quenching strength of 33 ns^{-1} at all three sites predicts that the triply-quenched complex falls on the experimentally measured Q₂ state. Horizontal lines show the experimentally measured parameters for Q₁ (orange) and Q₂ (red). b) Using this value for the OCP quenching rate (33 ns^{-1}) for all three sites, model results in brightness and lifetime are shown for five combinatoric conditions: (1) Three bound quenchers, $a'd'e$, shown in green. (2) Two bound quenchers at $a'd'$, shown in dark red. (3) Two bound quenchers at $a'e$ or $d'e$, which happen to

produce nearly indistinguishable photophysical states, shown in gray-blue, (4) a single bound quencher at a' or d' , which are symmetric and therefore produce degenerate photophysical states, shown in purple, and (5) a single bound quencher at e , shown in magenta. As described above in SI Note 4, the results predict that this scenario would produce 4-5 distinct quenched populations, rather than the two that were observed.

SI Note 4, SI p. 4

SI Note 4: Alternative combinations of binding sites and non-degenerate binding sites

In our Discussion, we present simulation results for two OCPs bound at the four pairs of rotationally symmetric compartments. We excluded non-rotationally symmetric combinations and more than two OCPs from consideration because these conditions would generally predict more than two quenched populations except under specific conditions. For example, we expect that three distinct populations would be observed for binding at two asymmetric sites: 1) the doubly-quenched population, 2) one of the single-quenching sites, and 3) the other single-quenching site. These would collapse into fewer than two populations only under special conditions, including cooperative binding of the two OCPs (most likely producing just one population) or if the two non-symmetric sites just happened to produce spectroscopically identical states (two populations).

For more than two sites (for example, for three bound) you would expect to see 1) three bound OCPs, 2) at least one (but likely more than one) quenching level that represents two bound OCPs and 3) at least one quenching level that represents one bound OCP (again, likely more than one). So, for three bound OCPs, we would expect to see at least three or more distinct populations. Again, this possibility cannot be completely eliminated because it is conceivable that strong cooperative binding or similar anomalous factors might prevent some of these combinatorically possible states from being populated. However, in our view the simplest possible explanation for seeing precisely two populations is that two OCPs can bind at symmetric sites.

To illustrate what happens if more than two OCPs are bound, we simulated a three-quencher scenario that includes two symmetric sites (therefore producing some degeneracy and lowering the number of expected states). We selected the quencher strength by requiring that the triply-quenched complex produce the Q2 photophysical parameters (SI Fig. S11a), and then used this quencher strength to simulate the various possible combinations of two and one quenchers (SI Fig. S11b). As described above, the results predict that this scenario would produce at 4-5 mutually distinguishable quenching populations.

Reviewer #3 (Remarks to the Author):

The manuscript by Squires and colleagues presents a single molecule spectroscopy study of the interaction between two isolated physiological players: OCP and PBS, which are involved in photoprotection in cyanobacterial cells. In its active form, OCP binds to a PBS and induces energy dissipation that is signified by the PBS' fluorescence quenching. Single molecule fluorescence spectroscopy experiments are performed in an ABEL trap, a system that allows tracking the fluorescing complexes in solution and record fluorescence brightness, fluorescence polarization, fluorescence lifetimes and fluorescence spectra. Here, the SMS allowed the authors to investigate some mechanistic details of the interaction between OCP and PBS. The discussion of the SMS results leads the authors to the conclusion that under saturating conditions more than one OCP molecule binds to a core of PBS.

While this idea is, as such, interesting and, given the symmetry of the PBS' cores, not impossible, the results and analysis presented in the manuscript do not convincingly support this hypothesis. Moreover, some results seem to argue against the proposed model of 2 OCPs binding to the PBS. Finally, the manuscript does not present sufficient controls allowing the readers/ reviewers to assess the impact of the ABEL trap condition on the interaction between OCP and PBS.

We appreciate Reviewer #3's careful reading of the manuscript, and suggestions to improve the work. However, we respectfully disagree with parts of the reviewer's assessment, and provide additional data and reasoning to substantiate our conclusions. To summarize the main points:

- a. The referee questions why previous bulk studies of OCP (population ratios, bulk lifetimes, bulk emission spectra) have failed to identify the two quenched states we observe. In fact, the previous work shows hints of more than one species present, as evidenced by the 1.2:1 binding ratio from Gwizdala et al. 2011. In fact, our single-molecule approach directly senses the differences between Q1 and Q2 that are hidden in ensemble averaging (see the clear pair of clusters in Fig. 3).
- b. The referee presents an alternative hypothesis that the less-quenched state we observe is not related to the presence of OCP. Our controls directly refute this possibility; neither Q1 nor Q2 is ever observed in the absence of OCP, regardless of excitation intensity. (see Figures S2 and S3)
- c. The referee questions why we never directly observe transitions between more quenched and less quenched states arising from different numbers of OCP present. Yet we easily see single complexes mostly showing Q2 at initial dilution to 10 pM, followed by less Q2 cases and more Q1 and ultimately unquenched complexes over time as the OCP molecules fall off. Transitions for a single complex, however, are rare under our conditions.

Together with the changes to the manuscript, we feel that these responses will clarify the few points in question, and serve to improve the manuscript. **Yellow highlights indicate responses that correspond to changes in the manuscript or SI.**

Previous in vitro characterization of the OCP-PBS system have shown that upon reisolation of the OCP-PBS complexes from the PBS exposed to an excess of OCP, each PBS bound, on average, 1.2 OCPs. While 1.2 is clearly higher than 1 suggesting that some PBS complexes could have attached more than 1 OCP (whether it is a specific binding or not, remains unclear), the spectroscopic properties of OCP-PBS complexes were always identical, also in two scenarios: 1) just after formation of a quenched complex under saturating conditions, and 2) after reisolation, which takes a few hours. Following the

interpretation from the submitted manuscript, after the re-isolation, the OCP-PBS complexes should be in state Q1, as shown by the OCP:PBS ratio of 1.2:1.

While the precise previous publication is not stated, we agree with the reviewer's point that the reported 1.2:1 ratio in a different bulk experiment (see bar chart and gel in the SI of Gwizdala 2011, our ref 29) strongly suggests that multiple OCPs were binding to some phycobilisomes in those studies. In fact, their exact language stated that they measured "1.2 OCP per PB varying from 1 to 1.5 OCP per PB depending on the sample". We would prefer to not over-interpret a single gel, especially in the absence of a standard curve calibration, signal saturation range of Commassie Brilliant Blue (CBB) staining, etc.

More importantly, in our experiment (lacking ensemble averaging), two clearly distinct quenching states were observed at saturating activated OCP, and as OCP molecules left the complex under extreme dilution, the one-OCP quenched state (Q1) was observed more and more frequently. It is important to note that the OCP-PB complex is relatively stable under our conditions, and OCP only dissociates over several hours even at 10 pM concentration. Critically, in our study we are working at ~pM concentrations for ABEL trap experiments, while typical stock concentrations for phycobilisomes are far larger. We do not find it surprising that previous studies at much higher concentrations observed highly stable complexes over several hours. Additionally, please note that OCP activation, binding, and purification were all performed with OCP in 40x excess and phycobilisomes at these higher concentrations, and dilution to trapping concentrations was only performed immediately prior to measurement. We have added language to our Methods section to further emphasize this point.

Methods, p. 9:

All sample prep was carried out at ~25 nM concentrations of the phycobilisome complex or greater, comparable to previous bulk studies. Immediately prior to trapping, samples were diluted in 0.8 M phosphate buffer, pH 7.5, with 1M sucrose, to a working concentration of approximately 10 pM.

However, just after the formation of quenched OCP-PBS complexes, they are expected to be in a spectroscopically different state Q2 (binding more OCPs), which is apparently not observed in ensemble fluorescence spectra published previously. It is unclear how a difference between Q1 and Q2 would be visible in SMS and not in ensemble.

We believe that this point highlights the power of single-molecule multivariate approaches at uncovering population-wide heterogeneity (see Gwizdala, Van Grondelle, Krüger, our Ref. 45). In bulk, the effects of contributing populations must be deconvolved. Therefore, unless an assumption was made in previous studies that two quenching populations existed, Q1 and Q2 would not have been previously discovered in bulk (especially given their photophysical similarity). It is only with our high-precision single-complex measurements that we were able to directly observe and differentiate them.

Moreover, as can be concluded from the fluorescence lifetimes (Tian et al 2011 and 2012) and following kinetic modelling (van Stokkum et al 2018) of OCP-related quenching, the presence of a second OCP in the PBS core would not change the efficiency of quenching, i.e. in ensemble experiments, whether one or two OCPs bind to the core, the outcome would be indistinguishable.

We directly adapted the van Stokkum 2018 model (which is based on the Tian 2011 and 2012 data) for this work, including the compartment connectivity and intercompartmental energy transfer rates. As detailed in our manuscript, we find that their model in fact clearly predicts that two OCPs produce a photophysical state that is different from that produced by a single OCP (Figure 5).

We expanded upon this model in our work by testing many different binding positions and combinations of binding positions for the OCP(s) on the phycobilisome, and found that many of these positions and combinations were also mutually distinguishable, as is discussed in our manuscript.

An insufficient characterization of the unquenched PBS complexes in the manuscript does not allow the reader to critically assess the results for the OCP-PBS. In Fig 2b, the authors suggest that PBS alone can only assume one photophysical state, namely the unquenched state (U). This result is inconsistent with the previous SMS reports for these complexes where excited PBS were reversibly switching between an unquenched and a quenched state (Gwizdala et al 2016 and 2018), with the switching rates increasing with the excitation intensity. In the example from Fig 2b no such switching or other-than-U states were shown.

It is good to compare to the previous SMS experiments on a surface, but we believe the Referee's conclusions about our solution-phase single-molecule work are in error. In Fig 2b, we present trapping data taken at low power so that the phycobilisome did not enter self-quenching states. This data serves to illustrate that the phycobilisomes are not structurally or electronically damaged by the ABEL trap, and that their observed photophysical parameters are indeed identical to those in bulk.

In our original SI (Figure S2), we presented data taken at even higher intensity (the same intensity that is used for our OCP-PBS measurements), where the phycobilisome exhibits self-quenching and blinking in a manner entirely consistent with that reported by Gwizdala et al (2016). In this data, many photophysical states are observed, but critically, none of these states reaches the level of quenching of Q1 or Q2. Therefore, neither Q1 nor Q2 could be caused by the phycobilisome alone, even under high excitation. This point is already discussed in the manuscript.

To provide more detail, we have added scatter plots to the SI from the high-power excitation of the unquenched phycobilisome, which clearly show that without OCP, neither Q1 nor Q2 is ever observed (note inset), even at high power (Figure S3).

Figure S3: Scatter density plots for unquenched phycobilisome at high excitation power

Scatter heatmap of the photophysical states (U_{damage}) observed for the unquenched CB-PB phycobilisome excited at high power (25 W/cm^2), shown in $Br-\tau$, $Br-\lambda_{\text{CM}}$, and $Br-FPol$ projections (left, right, and center, respectively). Each point represents 250 photons, colored according to the local density of points. The location of state B, very rarely observed here, is also indicated for reference. Despite the heterogeneity of photophysical states observed, Q1 and Q2 are not observed in this sample (see inset).

For a reason not explained in the manuscript, the experiments on the OCP-PBS complexes were performed using 3 times higher light intensity than for PBS alone, i.e., under conditions, in which the probability of finding PBS in the quenched states is higher.

We have added language to the Results section to further clarify the reason for the higher power employed for trapping experiments on the much dimmer quenched phycobilisomes:

Results, p. 3

A similar raw data trapping trace for several individual quenched OCP + CB-PB complexes can be seen in Fig. 2c (excitation: 15 W/cm²). Higher intensities were used to study quenched complexes because they are an order of magnitude (or more) dimmer than the unquenched phycobilisome. To achieve sufficient signal to noise, these higher powers were necessary. The trapped quenched complexes are clearly photophysically stable at these powers, as shown by the long, single-level trapping events (see Figures 2 and S3 for sample trapping events).

It is therefore possible that the Q1 state is related to light-induced dynamics of PBS and not to the activity of OCP. Moreover, the duration of different states presented in Fig 2c supports the idea that some of these states, in particular Q1, are indeed light-induced and that OCP is not present in these complexes (typically OCP forms very stable complexes with PBS). Following this thought, can it be excluded that only Q2 states would be OCP-related and Q1 states would represent intrinsic dynamics of PBS (induced by higher light intensity than in the control from Fig 2b). Moreover, the relative difference in fluorescence lifetimes between Q1 and Q2 (Q1 having ~2 times longer lifetime than Q2) is in agreement with the relative difference between the lifetimes of OCP-induced and light-induced (intrinsic, measured in absence of OCP) quenched states of PBS reported previously (Gwizdala et al 2018).

See above; in our solution-phase study, Q1 is never observed in the absence of OCP, even at high intensity. Moreover, Q1 is observed as a single, stable level, and is not part of fluctuating or blinking events, as evidenced from the raw data traces presented both in the manuscript (Figure 2) and SI (Figure S3). These results are inconsistent with Q1 being a transient blinking level of the phycobilisome.

The authors described that they did not observe a transition between Q1 and Q2 state. However, if 2 OCPs were to bind to a PBS it is far more likely that they would bind sequentially and not at the same time, i.e., first one OCP would bind and then the second – and the same for detachment. Since no such observation was reported it has to be assumed that the states Q1 and Q2 are accessed due to single events: for Q1, light-induced change of properties of a pigment in PBS; for Q2, binding of OCP. Assuming the two OCP scenario as proposed by the authors, transitions between the Q1 and Q2 states would be frequent due to the sequential attachment of OCPs to the PBS.

We do currently highlight the long-time-scale transition from Q2 to Q1 to unquenched states as OCP unbinds from the phycobilisome under dilute conditions (Figure 4).

For the case of transitions during the time we observe one trapped single complex, we have added a note to the SI that expands upon and clarifies, as follows:

SI Note, p. 3 of Supplementary Information:

SI Note 3: Transitions among quenched and unquenched states of the phycobilisome are infrequently observed in the ABEL trap

In this work we do not claim to definitively identify single-molecule transitions among the various states we have identified. This is for three main reasons:

1. Regarding the expectation of observing transitions: Our time-dependent unbinding data (Figure 4) indicates that the PB-OCP complex is relatively stable, and OCP unbinds over the course of hours. Here, we trap each complex for no more than a few seconds. Therefore, we expect that we would only rarely observe transitions during trapping events.
2. We do occasionally observe trapped objects that appear to change their state (a couple of examples of this are evident in the extended raw data set presented in SI Figure S3). However, these events are rare. This is consistent with the expectation described above for a stable PB+OCP complex, but it is also possible that those rare transitions could be replacement events in the ABEL trap, where a second complex (which might be in a different state from the trapped object) enters the trapping area and by chance replaces the initially trapped object. While we work at very low concentration to minimize this possibility, it cannot be discounted in the case of rare events. We therefore do not speculate in the manuscript as to the nature of these events.
3. Moreover, the active observation of binding or unbinding of single OCP to a single phycobilisome is more relevant to a study about binding kinetics than to the photophysical states present in the quenched complex, which is the topic of this work.

Finally, the manuscript does not test the proposed model against its biological relevance. In the cells the amount of OCP is smaller than that of PBS. This fact is included in the models of Tian et al. 2011 showing that even at low concentrations in the cells, OCP creates an efficient channel for energy dissipation from PBS, in effect providing photoprotection. If however, the number of PBS that bind OCP decreases by half (since every PBS can bind 2 OCPs), it is doubtful that OCP would still serve a photoprotective function in the cells.

Our results agree with those of Tian et al. (2011), and additionally demonstrate that while even one bound OCP can efficiently quench a phycobilisome, two bound OCPs quench it even more. We agree that this may prove to have interesting implications for the *in vivo* regulation of quenching by OCP. However, the ability to bind two OCPs certainly does not necessarily mean that every PB must always have two OCPs bound *in vivo*. Rather, as discussed in our conclusions, it may provide an additional knob for control over the degree of quenching or possibly photoprotective redundancy. It remains unclear how PBs and OCP are heterogeneously located in the cell and how PBs are energetically coupled and structurally interact with each other. This is a topic that will certainly merit future investigation, but that we feel is beyond the scope of the current work.

As summarized above the manuscript does not convincingly proof the existence of two quenched states induced by one or two OCPs. Instead, it may show the difference between light-induced and OCP-induced dynamics, a topic previously addressed in SMS studies of the interaction between OCP and PBS.

We respectfully disagree; please see above. Our detailed measurements show that it is not high intensity that generates the additional quenched state.

On top of that, the manuscript shows a few minor weaknesses that should be fixed:

- 1) *The manuscript does not address the possibility that the electric field present in the ABEL-trap could impact the: 1) photoactivation of OCP, 2) binding of OCP to PBS, 3) electronic interaction between the OCP and PBS, and 4) the physicochemical properties of PBS. Relevant controls are missing.*

These are all important concerns, and we thank the reviewer for bringing them to our attention. In order:

- 1) OCP is pre-activated and pre-bound to the phycobilisome prior to detection in the ABEL trap; therefore the fields of the ABEL trap are not relevant to the process of photoactivation or binding of OCP to PBs. **We have added language to the Results section to clarify:**

Results, p. 3:

Here, we examined both single CB-PB phycobilisomes and single pre-activated and pre-bound OCP + CB-PB complexes in the ABEL trap. A raw data trace in these variables for several consecutively trapped single unquenched CB-PBs is shown...

- 2) See #1.
- 3) In the trap, events from quenched complexes are photostable and many seconds in duration. Unquenched complexes (at lower illumination powers) are also photostable. Moreover, the lifetime and emission spectrum of the unquenched complex is identical to bulk. Therefore, it is evident that the ABEL trap does not substantively alter the photophysical properties of the phycobilisome upon trapping. Additionally, the topic of sample perturbation has been addressed in detail in several previous ABEL trap publications that are cited in this work (see for example Cohen and Moerner 2007 *PNAS*, Wang and Moerner 2014 *Nature Methods*, Squires and Moerner 2017 *PNAS*, and Yang and Moerner 2018 *NanoLett*). In all of these cases, which include some binding studies on proteins and on DNA, single-molecule samples behaved indistinguishably in the ABEL trap and in bulk / free solution.
- 4) See #3.

Points #3 and 4 are currently addressed in the Results section on p. 3, but **we have added an SI Note for additional clarification:**

SI Note 2: The electric fields of the ABEL trap do not alter the photophysical properties of analyte molecules

In the trap, events from quenched complexes are photostable and many seconds in duration. Unquenched complexes (at lower illumination powers) are also photostable. Moreover, the lifetime and emission spectrum of the unquenched complex is identical to bulk. Therefore, it is evident that the ABEL trap does not substantively alter the photophysical properties of the phycobilisome upon trapping. Additionally, the topic of sample perturbation has been addressed in several previous ABEL trap studies – we find that the rotational diffusion,¹ binding and unbinding kinetics,² structural conformation of biopolymers,³ and fluorescence parameters⁴⁻⁶ are unchanged from their bulk values.

- 2) *Regarding the characterization of the PBS in the absence of OCP, the authors did not explain why detached C-PC rods are more abundant in quenched complexes than in PBS alone. Since OCP does not directly interact with C-PC, this suggests that the conditions in OCP-related experiments had a larger impact on the PBS.*

This is an interesting question, and indeed was an observation that surprised us. One possible explanation, as noted by the reviewer, is that in the presence of OCP, C-PC and the PB core (APC) are somehow energetically (but not structurally) decoupled, allowing an energetic re-coupling upon the unbinding of OCP. However, because the quenched complexes are even dimmer than the C-PC rods, while unquenched complexes are much brighter than the rods, we cannot exclude the possibility that the higher proportion of trapped rods in the presence of OCP is related to the relative probability of trapping objects with different brightnesses. Put another way, we know that brighter objects are more readily trapped, and so these might be a bias towards trapping full PBs (rather than rods) in that sample, while that bias is lessened or even leans toward trapping rods in the much dimmer quenched sample.

For this reason, we do not speculate in this work whether or not OCP affects the stability of rods on the phycobilisome, but we agree that it is a very interesting question for future studies.

- 3) *Description of methods is not sufficient. E.g. it is not explained how, when and under which conditions, the photoactivation of OCP was induced.*

We thank the reviewer for this feedback. We have expanded our methods section to include more detail regarding the photoactivation of OCP.

Methods, p. 9

...This mixture was illuminated with 2000 $\mu\text{mol photons m}^{-2}\cdot\text{s}^{-1}$ of white light for 10 min at 23 °C, after which the resulting OCP-PB complexes were purified using a 100 kDa filter (Millipore Amicon Centrifugal Filters, Billerica, MA). Bound, purified samples were resuspended into 0.8 M phosphate buffer, pH 7.5, with 1 M sucrose, flash-frozen and stored at -80 °C until use. All sample prep was carried out at ~25 nM concentrations of the phycobilisome complex or greater, comparable to previous bulk studies.

- 4) *SMS figures show polarization measurements that are not discussed in the text.*

The polarization measurements serve primarily to verify that the dim complexes we measure undergo substantial depolarization by energy transfer, which is an additional indicator that those dim events represent quenched phycobilisomes. We have added some language to the results section to address the significance of the polarization measurements:

Results, p. 4

As expected from the time-resolved data, one dominant state (U) is observed for the unquenched CB-PB. At least three possible states are observed for quenched CB-PB + OCP: two that are dim with short lifetimes (states Q₁ and Q₂; see inset for the *Br*- τ projection in the left panel of Fig. 3b, and a third state (state B) that is dim and blue-shifted with an unchanged lifetime relative to the unquenched CB-PB. Both the quenched and unquenched phycobilisomes exhibit nearly complete depolarization (FPol \approx 0), indicating that substantial energy transfer among many non-aligned dipoles occurs. The expected values and standard deviations for each of these states were determined for each parameter via Gaussian fits (see SI Fig. S4-S7), and are presented in Table 1.

- 5) *It is not sufficiently explained why the authors used the “CB-PB” complexes instead of PBS isolated from a WT.*

The CB-PB mutant was used due to its simpler structure relative to WT phycobilisomes (Ughy and Ajlani 2004), which reduces the possibility of phycobilisome structural variations contributing to our data. We have updated the language in the introduction to more clearly reflect this point.

Introduction, p. 2

The two discs at counterparallel ends of each of the two bottom cylinders together contain three subunit variants (ApcD, ApcE, and ApcF), which together form the site of energy transfer to PSI and PSII *in vivo* via a red-shifted pigment emitting at 680 nm located in the ApcD/ApcE subunits, respectively^{48, 49}. Excitons are funneled into the PB core by six C-phycoerythrin rods⁵⁰, which in the wild-type *Synechocystis* PCC 6803 each consist of three stacked C-phycoerythrin hexamers. Here, we employ the “truncated” CB-PB mutant of *Synechocystis* PCC 6803⁵¹ to ensure structural consistency (see Methods for details). The literature consistently reports that CB-PB is effectively quenched by OCP^{29,33}. This mutant is useful because it has been previously suggested that both the architecture and the spectroscopic properties of wild type phycobilisomes, especially the C-phycoerythrin rods, may be dynamic and/or modular^{47,52-54}, facilitating structural remodeling *in vivo*^{54,55}.

- 6) *I would recommend not using the center of mass approach to analyze the spectra that are as asymmetrical as the spectra of PBS.*

We respectfully disagree with the reviewer on this point. In low-signal-to-noise-ratio situations, the center of mass remains a reliable statistical reporter of changes in distribution position as long as the extent of the distribution is well-defined to limit the influence of noise from positions outside of the distribution. It can also report on changes in distribution shape that are asymmetric with respect to the CM. We do agree that by reducing the dimensionality of the data, the CM could in principle fail to capture effects of multiple contributing spectra or large changes in the spectral distribution shape. However, it is evident from our data that CM works effectively in its role as a separation parameter to describe spectroscopic differences among the observed states. We therefore have chosen not to alter our approach in this analysis for the purposes of this manuscript, and we note that full spectra for each population from the single-molecule data have been provided both in raw and aggregate forms (Figures 2, 3, S2, and S9).

- 7) *In the introduction, when describing the core of PBS the “cylinders” are referred to as “rods”, which is highly confusing, given that it is C-PC and not APC that forms rods. In the discussion the cylinders are referred to as cylinders.*

We thank the reviewer for noting this inconsistency; we agree that “cylinders” is the more appropriate term. **We have altered the language in the introduction accordingly.**

Introduction, p. 2

The PB architecture (Fig. 1c) is assembled primarily from α - and β -subunits of phycobiliproteins with high mutual structural and chemical similarity⁴⁷. In *Synechocystis* PCC 6803, the PB core comprises three **cylinders** that each contain four allophycocyanin (Apc) disc-shaped $\alpha\beta$ trimers, most of which emit at 660 nm. The two discs at counterparallel ends of each of the two bottom **cylinders** together contain three subunit variants (ApcD, ApcE, and ApcF), which together form...

Reviewers' Comments:

Reviewer #1:

Remarks to the Author:

The authors have adequately addressed my queries/suggestions.

Reviewer #2:

Remarks to the Author:

The authors have discussed my previous questions very well.

The paper can be published.

Thomas Renger

Reviewer #3:

Remarks to the Author:

I am perfectly happy with the current version of this manuscript